# A cross-study transcriptional patient map of heart failure defines conserved multicellular coordination in cardiac remodeling

Jan D. Lanzer [1,2,5], Ricardo O. Ramirez Flores [1,3,5], José Liñares Blanco[3], Marco Steier [2,4], Ashraf Y. Rangrez [2,4], Norbert Frey[2,4] & Julio Saez-Rodriguez[1,2,3] ✉

Impaired cardiac function in heart failure (HF) involves tissue remodeling through multicellular coordination. Although individual bulk and single-nucleus transcriptomics studies have offered insights, they have not been integrated to study conserved tissue-wide responses, limiting understanding of multicellular processes in cardiac remodeling necessary for therapeutic translation. Here, we integrate 25 studies of bulk and single-nucleus transcriptomics, spanning 1524 individuals, to define consensus multicellular programs associated with HF. These programs reveal conserved fibrotic, inflammatory, metabolic, and hypertrophic processes, with fibroblast activity consistently predicting cardiomyocyte stress. Our integration revealed that multicellular programs in HF are largely independent of tissue composition, and that fibroblast activation reflects a broad phenotypic shift rather than solely the emergence of discrete subtypes. Projecting external datasets onto these programs showed that clinical recovery aligns with reversion of disease-associated programs. This integrative analysis across studies establishes a public reference for the exploration of multicellular coordination in HF.

Molecular descriptions of the inter- and intra-cellular signaling mechanisms of cardiac cells in health and disease are essential to describe the pathophysiology of heart failure (HF). Historically, bulk transcriptomics has been used to compare general functional and compositional characteristics of the healthy and failing hearts. However, technological advances now facilitate the profiling of single cells in tissues, expanding our knowledge on the specific molecular processes carried out by the muscular, vasculature, stromal, neuronal, and immune cells that compose the heart.

Increasing collections of single-cell atlases chart the compositional and functional diversity of cardiac tissue of distinct heart compartments, and during development or disease[1–10]. Each study has established its individual classification of single cells based on lineage (i.e., cell-types) and functional potential (i.e., cell-states), complicating the comparison of the behaviors of cardiac cells across distinct biological or clinical contexts. While efforts to establish unified cell ontologies[11], integrate studies[12], or provide access to the information from distinct single-cell studies[13] exist, they are focused on the generation of cell-taxonomies that lack the contextualization of single-cell behaviors in terms of multicellular processes[14,15].

Understanding multicellular cooperation is critical because it ensures proper functioning of the heart, and to distinct pathological processes that trigger similar multicellular responses in patients is not known. Emerging computational methods that leverage group latent variable models enable the inference of multicellular programs (MCPs) —structured patterns of coordinated gene expression changes across

¹Heidelberg University, Faculty of Medicine, and Heidelberg University Hospital, Institute for Computational Biomedicine, Heidelberg, Germany. ²DZHK (German Centre for Cardiovascular Research), partner site Heidelberg/Mannheim, Heidelberg, Germany. ³European Molecular Biology Laboratory, European Bioinformatics Institute (EMBL-EBI), Hinxton, Cambridgeshire, UK. ⁴Department of Cardiology, Angiology and Pneumology, Internal Medicine III, University Hospital Heidelberg, Heidelberg, Germany. ⁵These authors contributed equally: Jan D. Lanzer, Ricardo O. Ramirez Flores. ✉e-mail: saezlab@ebi.ac.uk

multiple cell types[14]. Unlike traditional single-cell analyses that focus on individual cell types, MCPs capture both cell-type-specific and multicellular processes simultaneously, offering a broader systems-level perspective. Given the availability of single-cell data from cardiomyopathies, applying multicellular integration methods presents an opportunity to move beyond cell-type classification and incorporate a tissue-centric perspective. Moreover, this approach allows for the reinterpretation of bulk transcriptomic data beyond cellular composition, complementing single-cell studies that often have limited patient cohorts. Thus, multicellular integration offers a powerful framework to study multicellular processes in heart disease using the available omics datasets.

We recently reported a consensus transcriptional disease signature of HF built from the meta-analysis of public bulk transcriptomics datasets that pointed to a convergent transcriptional response of ischemic (ICM) and dilated (DCM) cardiomyopathies[16]. However, given the resolution of bulk transcriptomics, we were unable to quantify to what extent the convergent disease signature of HF was the product of specific cell types or tissue changes in composition and multicellular coordination.

Here we created a multicellular reference of the HF transcriptome by integrating the single-cell and bulk transcriptional profiles from cardiac tissues of control and HF patients across 25 core studies (1524 patients) and nine supporting studies (24 mice samples and 667 patients) to provide consensus multicellular processes associated with inflammation, fibrosis, cardiac remodeling, and fetal reprogramming in distinct etiologies. First, we evaluated the consistency of the multicellular, molecular, and compositional changes of cardiac tissue during HF from single-cell studies. Then, we built a consensus patient map, a latent space that captures the variability of tissue samples across single-cell studies based on coordinated transcriptional changes occurring in multiple cell types, here referred to as MCPs. We reconstructed networks of cell-dependencies underlying multicellular coordinated responses upon HF and identified a central role of fibroblasts in predicting the behavior of the other cell-types, particularly stressed cardiomyocytes. We further studied the distribution of HF responses across the population of fibroblasts to link the diversity of molecular phenotypes to tissue-level responses. Moreover, we showed that the fibrotic MCPs of HF can be traced in bulk transcriptomics data. Finally, we highlighted the clinical relevance of our inferred consensus MCPs of HF by projecting independent single-cell datasets onto our patient map. We summarized our integration work and made it publicly available at Reference of the HF Transcriptome (ReHeaT, https://saezlab.shinyapps.io/reheat2/) to allow researchers to query if individual genes are associated with HF in a cell-type-specific matter, facilitating a more nuanced exploration of the disease at the multicellular level.

## Results
### Study curation for the creation of the reference of the HF transcriptome
We queried public repositories of single-cell and bulk transcriptomics data to identify studies that profiled left-ventricle tissue samples of healthy non-failing (NF) controls and HF patients with the same inclusion criteria to ensure consistency with our previous bulk resource (Methods, Fig. 1A, Supplementary File 1). In addition to the previously reported 16 bulk studies, we identified nine new HF studies —four single-nucleus (SN)[4,7–9] and five bulk studies[17–21]—that fulfilled inclusion criteria, resulting in 25 core studies. Nine HF studies that could not be included due to reasons including disease etiology, species, or sample size requirements (Methods) were termed supporting studies and were used to corroborate and reinterpret our findings (Table 1). To facilitate the comparison of all studies we aligned meta data and cell type annotations, by mapping the original labels provided by each study to a harmonized vocabulary covering seven major cell-

types: Cardiomyocytes (CM), fibroblasts (Fib), pericytes (PC), and endothelial (Endo), vascular smooth muscle (vSMCs), myeloid, and lymphoid cells. This mapping was performed using regular expressions to standardize cell-type naming across datasets. Rather than performing full single-cell integration, we opted for ontology-based harmonization, which ensures biologically meaningful alignment of broad cell types that is sufficient for dataset comparability and MCP inference[22]. This data curation resulted in the largest collection of transcriptomic HF tissue samples to date, spanning a total of 2215 samples (1524 tissue samples [SN 132, bulk 1392] from core studies [Fig. 1B], and 691 tissue samples from supporting studies).

### Clinical presentation of 24 core HF studies
To assess the clinical diversity of the distinct patient cohorts included in our curation, we compared core studies based on reported patient-level metadata. HF and NF patients were similar in age, with a mean of 51.7 years for HF and 52.1 years for NF (Fig. 1C, Supplementary Fig. 1A). Studies undersampled female patients, with a total mean of 27% for HF and 44% for NF, though recent single-nucleus studies showed a higher proportion of female patients (bulk studies: HF 24%, NF 42%; SN studies: HF 40%, NF 51%) (Fig. 1D, Supplementary Fig. 1B). DCM was the most common etiology across studies, accounting for 60% of HF patients (59% in bulk, 65% in SN studies), followed by ICM at 31% (32% bulk, 11% SN) and HCM at 4% (3% bulk, 23% SN) (Fig. 1E, Supplementary Fig. 1C). The left ventricular (LV) free wall was the most sampled location (73% of total HF patients, 74% bulk, 58% SN), with the LV apex less frequently sampled (10% of total HF patients, 7% bulk, 42% SN) (Fig. 1F). No septal samples were included in the collection. HF samples were acquired after cardiectomy (83% of total HF patients, 85% in bulk, 59% in SN) or during left ventricular assist device (LVAD) implantation (16% total, 15% bulk, 22% SN) (Fig. 1G, Supplementary Fig. 1D). Cardiomyopathies can arise from primary (familial) or secondary (acquired) factors. SN studies reported 60% of DCM and HCM patients (and one arrhythmogenic right ventricular cardiomyopathy patient) were suspected to have a primary etiology (Fig. 1H, Supplementary Fig. 1E). Left ventricular ejection fraction (LVEF) and body mass index (BMI) were documented in three of the four SN studies (Fig. 1I), with a median BMI of 27 kg/m² for NF patients and 26 kg/m² for HF patients, and a median LVEF of 60% in NF and 20% in HF patients. Notably, race was reported for only 52% of SN patient samples, with 42% of those identified as white, 8% as African American, and less than 1% as Asian or South Asian (Supplementary Fig. 1F). In summary, our curation reveals notable sampling biases in gender, age, race, and disease stage, likely due to challenges in cardiac biopsy acquisition.

### Comparison of the core HF studies in bulk and single cell for molecular convergence
First, we compared the molecular similarity of HF in the curated core bulk and single-nucleus transcriptomics studies. These comparisons focused on three tissue characteristics: bulk expression profiles from bulk transcriptomics data, which represent the overall molecular state of the tissue; cell-type compositions from single-nucleus transcriptomics, which describe the structure of the profiled samples; and MCPs, which capture coordinated molecular activities across different cell types within the tissue. These MCPs represent interactions and shared molecular activities between various cell types, helping to define the functional states of the tissue. We hypothesized that if two independent studies using the same technology captured similar molecular processes in HF, a disease classifier trained on one study's data could predict the disease status of heart tissue samples in the other (Fig. 2A). This approach allowed us to assess how well independent studies with different technical (e.g. number of profiled cells, number of genes and sequencing depth) and clinical characteristics (Fig. 1, Supplementary Fig. 1) aligned at various scales.

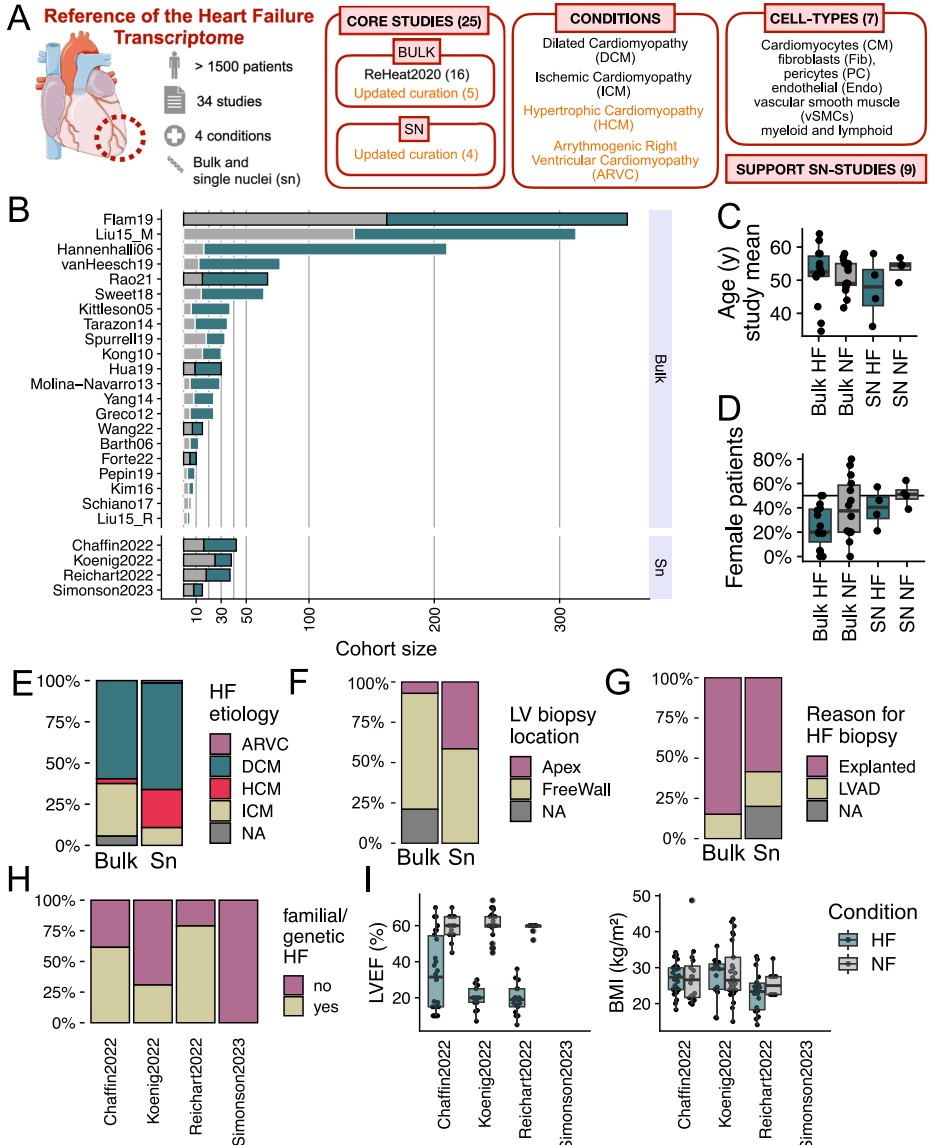

**Fig. 1 | Study and data overview and metadata presentation. A** Our updated reference integrates the molecular profiling of heart tissues of patients with HF and control donors from bulk and single-nucleus (SN) technologies, across distinct end-stage conditions, and cell types. In addition, we collected supporting data from studies profiling related heart conditions, smaller sample sizes, and mice models. **B** Sample size (x-axis) of core HF studies separated for bulk and SN technologies. Control patients (NF) in gray and HF (HF) patients in green; the black outline indicates new data added to the reference. **C** Distribution of mean patient age per study, categorized by sequencing modality and HF status. **D** Percentage of female patients per study, categorized by sequencing modality and HF status. **E–G** HF patient samples across bulk and SN studies categorized by (**E**) HF etiology, **F** sampled location, and **G** reason for biopsy. **H** HF patient samples in SN studies categorized by genetic or familial cardiomyopathy. **I** Distribution of left ventricular ejection fraction (LVEF, left) and body mass index (BMI, right) across heart tissue samples in SN studies. Boxplots in (**C**, **D**) display the minimum, first quartile (Q1), median, third quartile (Q3), and maximum; outliers lie beyond 1.5 times the interquartile range (IQR) from Q1 or Q3 and are shown as points. Source data are provided as a Source Data file. Heart icon in A by Servier https://smart.servier. com/ is licensed under CC-BY 3.0 Unported https://creativecommons.org/licenses/by/3.0/.

## Comparison and meta-analysis of the core bulk transcriptomics studies

We compared the bulk studies in a pairwise manner to assess their agreement on gene expression changes reported in HF (methods) and we found that the overlap of differentially expressed genes of the new studies was low (mean Jaccard index 0.055, Supplementary Fig. 2A). This low overlap aligned with our previous work[16] showing that variation in cohort size and non-probabilistic sampling design affects the statistical power of differential expression testing and the rankings of $p$ values and $t$ statistics. However, cross-study enrichment analysis showed that upregulated genes in one study tended to be enriched at the top of the gene-level ranking (mean enrichment score = 0.58),

while downregulated genes were enriched at the bottom of another study (mean enrichment score = −0.54, Supplementary Fig. 2B). This indicated that despite the low overlap in differentially expressed genes, their transcriptional directionality was conserved. Consistent with this, pairwise study classification yielded high accuracy (mean AUROC = 0.89, Fig. 2B; Methods), confirming that studies captured the same transcriptional patterns in HF, in agreement with our earlier findings[16]. These results supported the inclusion of new studies to refine the consensus signature through a gene-level meta-analysis (Fig. 2C). Additional analysis of the updated consensus signature, including comparisons with the previous version in terms of gene set overlap and classifier performance, showed that its predictive value for

**Table 1 | Overview of data sets used in this study**

| StudyID | Data modality | Cohort | Samples | HF etiology/model | Species | Citation |
|---|---|---|---|---|---|---|
| Wang22 | Bulk | Core Study | 15 | ICM | Human | 18 |
| Forte21 | Bulk | Core Study | 10 | ICM | Human | 20,21 |
| Rao21 | Bulk | Core Study | 67 | HF | Human | 19 |
| Hua19 | Bulk | Core Study | 30 | ICM | Human | 21 |
| Flam19 | Bulk | Core Study | 354 | HCM, DCM | Human | 17 |
| Chaffin2022 | Single-nucleus | Core Study | 42 | HCM, DCM | Human | 8 |
| Koenig2022 | Single-nucleus | Core Study | 38 | DCM | Human | 7 |
| Reichart2022 | Single-nucleus | Core Study | 37 | DCM, ARVC | Human | 4 |
| Simonson2023 | Single-nucleus | Core Study | 15 | ICM | Human | 9 |
| Hill2022 | Single-nucleus | Supporting Study | 12 | CHD | Human | 10 |
| Nicin2022 | Single-nucleus | Supporting Study | 5 | HCM | Human | 78 |
| Liu2022 | Single-nucleus | Supporting Study | 15 | Sarcoidosis, ICM | Human | 79 |
| Kuppe2022 | Single-nucleus | Supporting Study | 20 | MI | Human | 5 |
| Amrute2023 | Single-nucleus | Supporting Study | 13 | LVAD | Human | 6 |
| Mehdiabadi2022 | Single-nucleus | Supporting Study | 10 | pediatric DCM, fetal | Human | 80 |
| Litviňuková2020 | Single-nucleus | Supporting Study | 14 | Healthy | Human | 1 |
| McLellan2020 | Single-cell | Supporting Study | 8 | AngII | Mouse | 81 |
| Ren2020 | Single-cell | Supporting Study | 16 | TAC | Mouse | 82 |
| ReHeaT version 1 | Bulk | Core Studies | 916 | ICM, DCM | Human | 16,83–97 |

*ICM* ischemic cardiomyopathy, *HF* heart failure, *DCM* dilated cardiomyopathy, *HCM* hypertrophic cardiomyopathy, *ARVC* arrhythmogenic right ventricular cardiomyopathy, *CHD* congenital heart disease, *MI* myocardial infarction, *LVAD* left ventricular assist device, *AngII* angiotensin II, *TAC* transverse aortic constriction.

HF remained stable despite the reordering of the top genes (Supplementary Note 1, Supplementary Fig. 2C–F). Overall, these results confirmed the existence of a stable transcriptional response in HF from bulk transcriptomics.

**Comparison of the single-nucleus transcriptomics core studies**
We then compared single-nucleus studies by summarizing each individual atlas into a patient-level description of the molecular profiles for each cell type in our ontology, as well as their relative compositions (Fig. 2A). To enable cross-study comparisons, we generated pseudobulk expression profiles for each sample and cell type. As expected, we observed variation in the number of genes measured and the number of cells captured across studies, reflecting technical differences (Supplementary Fig. 3A–C, Supplementary Note 2). Additionally, we quantified the levels of gene expression contamination within each pseudobulk profile to account for the effects of ambient molecules, which were comparable across studies and could influence the estimation of cell-type-specific disease signatures (Supplementary Fig. 4A–D, Supplementary Note 2). Next, we evaluated the consistency of cell-type annotations across studies, after harmonizing them to a joint ontology, by comparing gene expression markers for each cell type calculated independently in each study. We observed agreement of markers between cells aligned to the same type (median Jaccard Index 0.48), indicating that the studies classified cells into comparable types. In contrast, comparisons between cells of different types showed very low agreement (median Jaccard Index 0.002), further supporting the accuracy of the original cell annotations (Supplementary Fig. 3D). Finally, we generated a consensus set of cell-type markers across studies by performing a meta-analysis of the differentially expressed genes for each cell type (Supplementary File 2).

Pathologic tissue remodeling often manifests as shifts in the abundance of resident cell lineages. To identify such a compositional disease signature of HF, we quantified and compared cell-type compositions across all tissue samples in the core SN studies (Fig. 2D). Based on the proportions of the seven cell types defined in our ontology, we observed modest grouping of patients by HF status (one-sided $t$ test, null = 0, adj. $p$ value = 0.01, 0.000005, for HF and NF patients, respectively) and no clear grouping by study (median silhouette width for HF, NF, and study = 0.14, 0.19, and −0.02, respectively, where higher values indicate well-defined clusters; Supplementary Fig. 5A). We next focused on how compositional differences between failing and non-failing hearts varied across the core studies. First, we inferred compositional disease signatures (i.e., differences in cell-type proportions between failing and non-failing hearts) for each study independently using differential compositional analysis across cell-types (Fig. 2E). We then tested how well these signatures classified failing and non-failing hearts in the other core studies, and we summarized these results using receiver operating characteristic (ROC) curves (Fig. 2F). Pairwise classification tests showed an average area under the ROC curve (AUROC) of 0.86, suggesting that the direction of cell-type compositional changes was comparable across studies. Finally, to identify the most consistent compositional changes in HF across studies, we used linear mixed models of cell-type proportions using as predictors the HF status of a patient while controlling for individual studies treated as random effects. We observed consistent patterns of cell-composition changes across studies, despite low effect sizes and high variability between studies (Fig. 2E). All cell-types changed between failing and non-failing hearts except for fibroblasts (adj. $p$ value < 0.05, mean proportion of explained variance associated with study across cells = 0.22), with a decrease in CMs and PCs and an increase in lymphoid cells being the most characteristic changes in failing hearts. Although cell-type proportions varied between studies, the results consistently revealed specific compositional changes in HF. However, the imperfect separation of conditions suggests that, while these proportions offer some insight into disease status, they could be influenced by study-specific factors, such as tissue sampling design, and/or that composition varies biologically between patients independently of HF status.

Next, we explored whether the observed differences in tissue composition among patient groups influenced the comparability of the gene expression profiles of the cell types in tissue samples from failing and non-failing hearts across single-nucleus studies. We

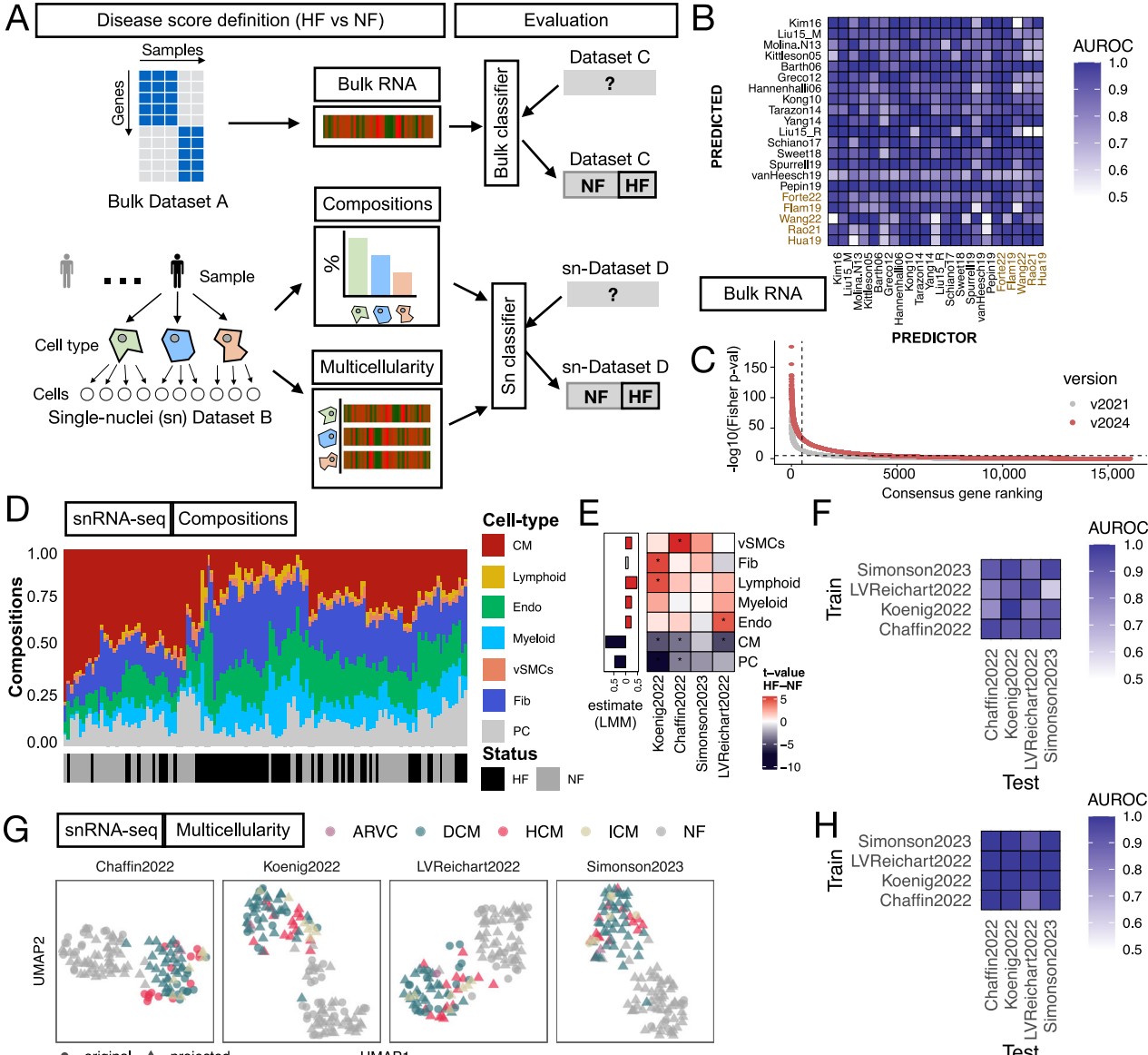

**Fig. 2 | Comparison of molecular descriptions of heart failure across studies.**
**A** Similarities of the molecular descriptions of HF between pairs of bulk or single-nucleus (SN) transcriptomics studies were approximated with the performance of disease classifiers built with each study of our core curation. The disease classifier from bulk studies used the changes in expression of each gene, while the classifiers of SN-studies were built with cell-type compositions or multicellular programs.
**B** Area under the receiver operating characteristic curve (AUROC) of pairwise predictions of disease classifiers built from all individual bulk studies included in our core collection using the top 500 differentially expressed genes. Red labels mark the study expansion. **C** Gene ranking of conservation of gene deregulation events in HF across bulk transcriptomics studies based on the adjusted *p* value of a Fisher combined analysis. The dotted line shows the top-500 genes. **D** Hierarchical clustering of all SN-transcriptomics tissue samples based on the composition of the seven major cell-types used in our ontology. **E** T-statistics (heatmap) and estimate of difference between failing and non-failing hearts from the differential compositional analysis using t-tests and linear mixed models, respectively. Stars denote an adj. *p* value < 0.05. **F** AUROC of pairwise predictions from disease classifiers built from all core SN studies using cell-type compositions. **G** Uniform Manifold Approximation and Projections (UMAP) of the multicellular programs describing the variability of the respective tissue samples of each core SN study. Tissue samples of the rest of the studies are projected into each latent space and distinguished with the shape of the dot. Colors highlight different HF etiologies. All plots include 132 patients. **H** AUROC of pairwise predictions from disease classifiers built from all core SN-studies using multicellular programs. In all panels Cardiomyocytes (CM), fibroblasts (Fib), pericytes (PC), endothelial (Endo), and vascular smooth muscle (vSMCs) cells. Heart failure (HF), and non-failing (NF) hearts. Source data are provided as a Source Data file.

reasoned that, since cells in tissues function as collectives, the molecular state of a tissue sample could be represented by coordinated transcriptional events across multiple cell types, where gene expression changes of one cell type relate to the changes of other cell types, referred to here as MCPs. To infer MCPs that describe patient variability within each study, we applied multicellular factor analysis[22]. Briefly, this method takes pseudobulk gene expression matrices from each cell type and infers a latent space that decomposes tissue sample variability. Each latent component defines an MCP, capturing coordinated gene expression changes across cell types. For each MCP, we obtained patient-specific activation scores and gene weights across cell types, enabling the interpretation of multicellular transcriptional states (Supplementary Fig. 5B). The MCPs of individual studies in the SN-core collection captured variance related to HF and other clinical covariates, clearly separating failing from non-failing hearts. These results suggested the presence of MCPs that reflect HF-associated

tissue remodeling within each study (Supplementary Note 3, Fig. 2G, Supplementary Fig. 5C–E).

When examining the individual cell types participating in these programs, fibroblasts showed the highest explained variance associated with HF (Supplementary Fig. 5C, mean explained variance = 25%, one-sided $t$ test $p$ value = 0.008), while lymphoid cells exhibited the lowest (Supplementary Fig. 5C, mean explained variance = 6%, one-sided $t$ test $p$ value = 0.003). These results contrast with our previous compositional analyses, where lymphoid cells showed significant compositional changes in HF, while fibroblast compositions remained stable across studies. This contrast highlights the complex interplay between compositional and molecular changes in failing heart tissues: although cell-type compositional shifts are associated with myocardial remodeling in HF, they do not necessarily align with the independent, coordinated molecular changes observed.

To compare the similarity of MCPs in predicting HF across studies, we projected the samples from the other three core studies into the multicellular space of each individual study and evaluated their ability to differentiate failing from non-failing hearts using pre-trained classifiers (Fig. 2G, H, Methods). The classification tests yielded a mean AUROC of 0.98, indicating high concordance in the molecular state of the tissues across studies. When assessing the contribution of individual cell types to the classification task (Methods, Supplementary Fig. 5E), we found that all cell types, except lymphoid cells, had a mean AUROC of 0.85 or higher. This suggests that most cell types contribute equally to the MCPs underlying HF. These findings suggest that, despite technical and clinical differences among the core SN studies, these studies share common multicellular processes that distinguish failing and non-failing hearts.

## Multicellular patient map of end-stage HF

Given the observed multicellular molecular similarity across the core SN studies, we built a joint multicellular patient map using multicellular factor analysis to identify both shared and specific axes of variation in the tissue samples (Fig. 3A). We decomposed the variability in gene expression across cell-types and patients into 10 MCPs, with a median R² across cell types of 31% (Supplementary Fig. 6A–E). The multicellular space showed little to no variability associated with body mass index, age, or sex (mean $R^2$ of 0.3%, 3%, and 0%, respectively). Differential expression analysis testing for interactions between HF and sex or age confirmed no significant effects, with a mean of 1.29 and 0.1 genes, respectively, showing a significant interaction coefficient across cell-types (Supplementary Fig. 6F). However, 17% of the gene expression variability across cell types was associated with HF (Supplementary Fig. 6D). Among all cell types, fibroblasts exhibited the highest percentage of variance explained by HF, with an average of 24.5% across studies ($t$ test $p$ value = 0.01), despite not having the largest number of genes in the model (Supplementary Fig. 6A). This suggests that fibroblasts undergo the most pronounced molecular changes in response to HF and that multicellular coordination of other cell-types.

Within our model, we identified two major multicellular programs (MCP1 and MCP2) that describe the coordinated multicellular differences between non-failing and failing hearts (Fig. 3B). MCP1 explained differences consistently across all studies (ANOVA adj. $p$ value < 0.05), while MCP2 separated conditions only in Reichart2022 and Koenig2022, indicating that our model is able to capture via the different programs both convergent and study-specific multicellular responses in HF (Supplementary Fig. 6E). However, we also observed differences in the distribution of HF samples across MCP1 and MCP2 that were associated with the study of origin (ANOVA adj. p value = 0.0245 and 0.0001 for MCP1 and MCP2, respectively). This suggests that, despite the generalizability of the multicellular HF programs, their variability is still influenced by technical factors. By fitting linear mixed models to both MCPs using patient information as predictors and accounting for

studies as random effects, we found no association between the distribution of HF patients across MCP1 and MCP2 and their organ sampling sites, acquisition mode, primary or secondary causes of HF, or etiology (Supplementary Fig. 6G). Differential expression testing across etiologies confirmed the low signal, with a mean of 8.5 and 37.1 differentially expressed genes for HCM and ICM, respectively, across cell types (Supplementary Fig. 6H, Supplementary File 3). Additionally, there was no correlation between the MCP1 and MCP2 scores of HF samples and the compositions of the seven major cell types analyzed, indicating that the activation of these MCPs is independent of tissue composition.

The MCPs capture coordinated dysregulation of gene expression across cell types and can be interpreted at the gene level to identify conserved markers of cell-type-specific deregulation in HF (Fig. 3C). As such, the MCPs provide a rich resource for generating hypotheses about stable disease markers, such as *PLCE1* in CMs[23] or *FKBP5* in Endothelial cells[24]. We observed that 46% and 49% of the genes associated with MCP1 and MCP2 (absolute weight > 0.1), respectively, were relevant to more than a single cell type, indicating that HF transcriptional changes reflect a combination of both cell-type-specific and multicellular processes.

To better interpret the two-dimensional HF patient map spanned by MCP1 and MCP2, we enriched prior knowledge gene sets that reflect cellular and molecular functions (Supplementary File 4), considering both cell-type-specific and multicellular perspectives (Fig. 3D). The multicellular response captured by MCP1 was associated with pathways related to hypertrophy and fibrosis, including calcineurin[25,26], histone deacetylases[27,28], vasoactive intestinal polypeptide[29,30] and interferon-α and -γ signaling[31]. In contrast, MCP2 was linked to inflammation, with pathways involving TNF-α, TGF-β, TNF/stress responses, and apoptosis. Many cell-type-specific pathways were shared between MCP1 and MCP2. In cardiomyocytes, upregulated genes were related to sarcomere organization and z-disc morphology, while genes associated with oxidative phosphorylation and fatty acid metabolism were downregulated. In fibroblasts, collagens were upregulated in both MCPs, while genes in the RECK pathway (including TIMPs and MMPs) were downregulated, indicating increased extracellular matrix (ECM) production and decreased degradation. Endothelial cells showed upregulation of genes related to cell differentiation, with MCP1 particularly enriched for interferon-α and -γ genes, while lymphangiogenesis-related genes were downregulated. Pericytes downregulated their responses to nitrogen stress in MCP1, and myeloid cells showed a weak association with JAK-STAT signaling in MCP2. These results highlight key processes in HF, such as fibrosis, vascularization, and cardiac remodeling, reflecting expected biological pathways in cardiac tissue. While we observed general convergence and conservation of molecular profiles across HF samples from different studies, the molecular patient map built from MCPs revealed variable expression levels and combinations of cellular processes across major cell types.

## Blueprint of multicellular processes and cell-cell communication in HF

MCPs represent global patterns of cellular coordination, allowing us to extract insights into cell dependencies by examining co-expression and ligand-receptor interactions across cell types. MCP1 was associated with HF in a larger proportion of patients while capturing more of the transcriptional variance than MCP2, and was selected to investigate these cell dependencies in greater detail. For this, we inferred from MCP1 two cell-cell coexpression networks, here referred to as multicellular coordination networks, with directed edges that describe to what extent the molecular profile of one cell-type could predict the profile of each other cell-type in both failing and non-failing hearts (Methods, Supplementary Fig. 7A, B, Fig. 3E, Supplementary Fig. 8A). We found a Spearman correlation of 0.68 ($p$ value = 1e-06) between the

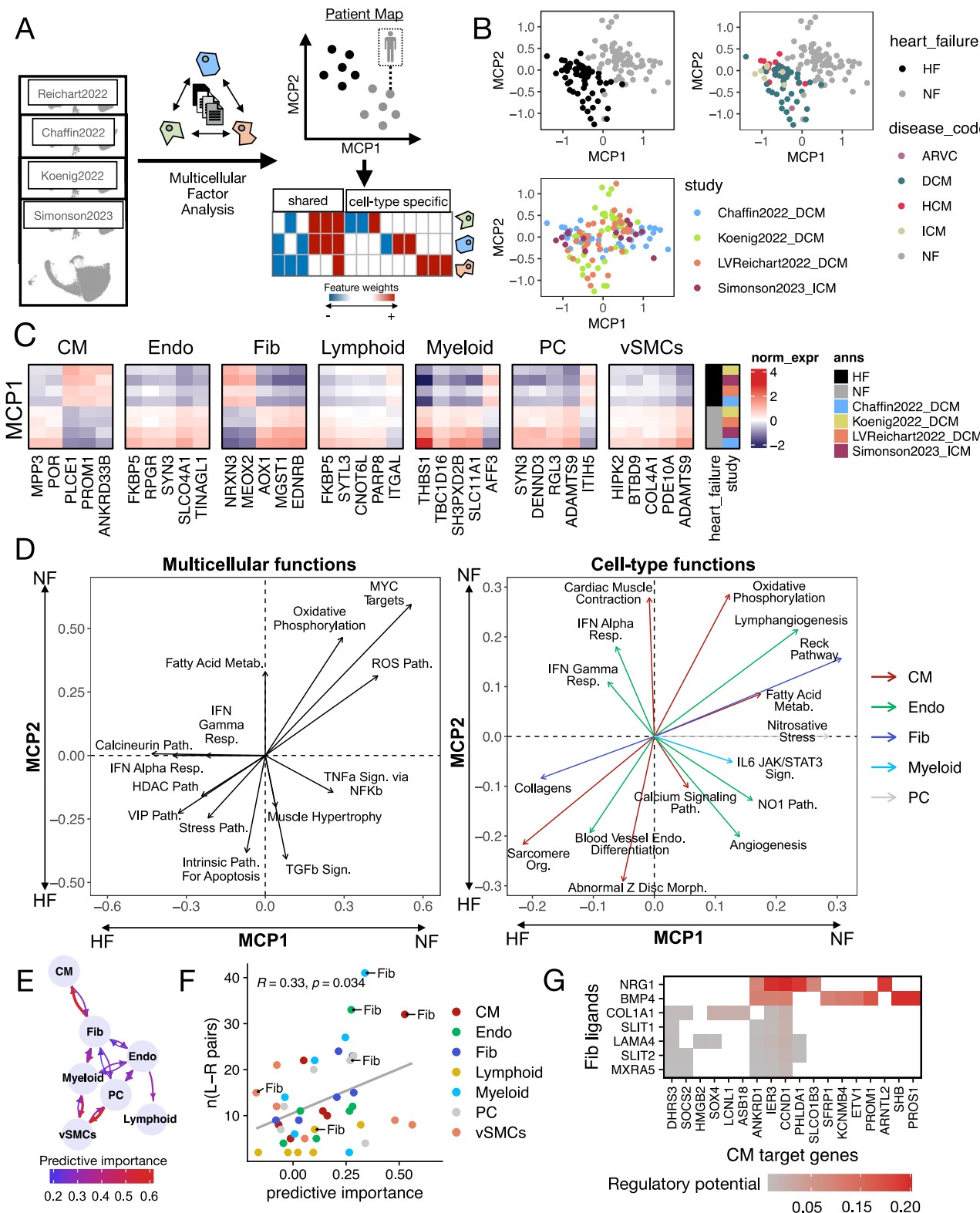

predictor importances of the two networks, suggesting that their overall organization is similar (Supplementary Fig. 8B). However, the strength of the predictor importances changed between failing and non-failing hearts (one sample *t* test of the difference of predictor importances per cell-type, with a null hypothesis of 0, adj. *p* value < 0.05), with cardiomyocytes' predictors showing the largest shift in importance, and fibroblasts taking on a greater predictive role. The failing heart co-expression network showed strong connectivity within

the vascular niche (involving endothelial cells, pericytes, vascular smooth muscle, and myeloid cells), while cardiomyocytes were primarily predicted by fibroblasts (Fig. 3E).

To investigate potential mechanisms of cell–cell communication underlying the failing-heart co-expression network captured by MCP1, we inferred ligand-receptor interactions across cell types (Supplementary Fig. 7C, Supplementary Fig. 8C). We observed a modest correlation between the number of potential ligand-receptor pairs in a cell

**Fig. 3 | Consensus multicellular landscape of human heart failure.**
**A** Multicellular factor analysis was used to integrate the patients' profiles across the single-nucleus core studies. The integrative model represents each sample in terms of latent variables, referred here as multicellular programs (MCP), that capture gene expression variability across cell types, patients, and studies. Each MCP can be understood as a collection of genes whose expression is coordinated across cell types. **B** Patient map built from MCP1 and MCP2 with samples ($n = 132$) colored by their disease status, etiology, and study of origin. **C** Mean standardized gene expression of the top 5 genes in MCP1 whose expression was captured in all studies. Data were grouped by disease status and study of origin. **D** Functional dissection of the patient map built from MCP1 and MCP2 from a multicellular (left) or cell-type (right) perspective. Each functional vector represents the level of enrichment of a function in the location of the map where the arrow points to. The larger the arrow, the more enriched the function. Gene sets were manually selected for representation from a set of gene sets enriched in either MCP1 and/or MCP2 (adj. two-sided $p$ value < 0.1, hypergeometric test). **E** Multicellular coordination network of HF processes captured by MCP1, where each arrow describes how important the expression of a given cell-type is to predict the expression profile of another one (Methods). Predictive importances come from linear mixed models of cell-type signatures of MCP1. Importances below 0.2 were not included. **F** Association between the predictive importance of a pair of cell-types (sender and target) and the number of potential ligand-receptor coexpression events. Pairs of cells are colored by their target cell-type and highlighted when fibroblasts (Fibs) are the sender cell-type. Pearson's correlation coefficient and its $p$ value is displayed. **G** Regulatory potential score, as estimated by NicheNet, represents the potential of fibroblasts' ligands in contributing to the regulation of cardiomyocyte genes (MCP1 gene loading < −0.2). In all panels, Cardiomyocytes (CM), fibroblasts (Fib), pericytes (PC), endothelial (Endo), vascular smooth muscle (vSMCs) cells. Heart failure (HF), and non-failing (NF) hearts. Source data are provided as a Source Data file.

type pair and their predictive importance in the multicellular coordination network (Spearman correlation = 0.33, $p$ value = 0.034, Fig. 3F, Supplementary Fig. 8D), suggesting that a minimal set of communication interactions could drive multicellular coordination (Fig. 3F). The highest number of ligand-receptor interactions in HF was observed between fibroblasts and cardiomyocytes, as well as between myeloid and endothelial cells, which had, in addition, high predictive weights in the multicellular coordination network. Given the strong coordination between fibroblasts and cardiomyocytes, the higher number of inferred ligand-receptor interactions, and the role of fibrosis in myocardial remodeling, we used NicheNet[32] to estimate the potential regulatory effect of ligands secreted by Fibs on the expression of HF-associated genes in CMs (Methods, Supplementary Fig. 7C). Many ligand-receptor pairs were mediated through ECM components, such as *COL1A1*, *COL3A1*, *LAMA4*, and *MXRA5*, supporting the idea that ECM remodeling influences CM transcription and phenotype, especially in regard to stress, stiffness, or hypertrophy[33]. Additionally, ligands like *BMP4* and *NRG1* were identified as potential regulators of the CM stress response within MCP1 (e.g., *ANKRD1*, *SACS*, *CCND1*) (Fig. 3G). *BMP4* is known to play a role in HF, particularly in cardiomyocyte transdifferentiation[34], while *NRG1* has been implicated in cardiomyocyte division and migration[35]. We corroborated experimentally in cell cultures of neonatal rat ventricular cardiomyocytes (NRVCM) the gene regulatory effects of *BMP4*, *NRG1* and *MXRA5* on CM target genes like canonical hypertrophy (*NPPA*, *NPPB*) and fibrosis (*TGFB1*, *COL1A1*) marker (Supplementary Fig. 9A, B), as well as on predicted target genes within MCP1 (e.g. *ANKRD1*, *CCND1*, *IFNAR2*, *NAV2*, *MCL1*, *SACS*, *SOCS2*) (Supplementary Fig. 9C, Supplementary Note 4). Thus, these ligands could be key regulators of coordinated multicellular fibrotic processes in HF and hence potential modulators of myocardial remodeling. In HF, fibroblasts were best predicted by CMs and Myeloids. The inference of potential ligands from these cell types regulating fibroblast gene expression (Supplementary Fig. 8E, F) identified a set of ligands, some of which have been previously linked to have pro-fibrotic effects, including *LGALS9*[36] and *PDCD1LG2*[37] from Myeloids and *CALR*[38] from CMs. In summary, we reconstructed a multicellular blueprint of cell-cell interactions in HF, revealing the coordinated processes captured by MCP1 and their potential ligand-receptor mechanisms (Supplementary File 5). Our analysis highlights the central coordinating role of fibroblasts in HF, particularly the increased communication between fibroblasts and cardiomyocytes, supported by predictive models of ligand-receptor interactions and cellular coordination, as well as by experimental validation of selected ligands in vitro.

### Division of labor in fibroblast during HF

Fibroblasts are highly versatile cells that perform a wide range of biological functions, including immune modulation and ECM remodeling. Due to their central role in regulating tissue remodeling,

fibroblasts have become a major focus of therapeutic interest in disease contexts[39–42]. Since our analyses indicated that fibroblasts coordinate multicellular responses associated with myocardial remodeling and the vasculature niche, we then investigated how the distinct functions of fibroblasts are distributed across the population of single cells to gain deeper insight into the multicellular coordination of HF processes of MCP1.

First, we defined common fibroblast cell-states by integrating 242,045 fibroblasts from core single-nucleus studies (Supplementary Fig. 10A–G). We identified six conserved fibroblast states via clustering (Fig. 4A), with all patient samples contributing cells to each state (Supplementary Fig. 10E). We characterized states by their expression of ECM components as well as pathway and cytokine activity signatures (Fig. 4B, Supplementary Fig. 10F–G, Supplementary Fig. 11A–D). Briefly, Fib0 (*COL4A1*+) expressed basement membrane components, suggesting a role in tissue homeostasis. Fib1 (*POSTN*+ and *THBS4*+), characterized by the expression of matrifibrocyte markers[43], was enriched for TGFβ signaling and core matrisome-related genes. Fib2 (*KAZN*+) exhibited an immune-related expression profile, including TNFα and interleukin signaling, and expressed secreted ECM factors. Fib3 (*PCOLCE2*+ and *SCARA5*+) was associated with secretory factors and angiogenesis, along with signatures of activity of BMP4. Interestingly, Fib4 and Fib5 were the only states that did not strongly express ECM-related genes. Fib4 was characterized by Hedgehog signaling, IL4, and BMP6 signatures of activity, while Fib5 displayed signatures of cytokines such as TWEAK and IL2. By analyzing the compositional changes of fibroblast states between failing and non-failing hearts, we observed a consistent pattern across studies. Fibroblast states Fib1 and Fib4 expanded, while Fib0, Fib2, and Fib3 decreased in abundance in HF (linear mixed model $p$ value < 0.05, Fig. 4C, Supplementary Fig. 11E). Our single-cell integration of fibroblasts highlights the diverse functional roles of this cell-type in the heart and suggests that upon HF, distinct populations associated with an increased ECM production and inflammatory responses are favored.

We next investigated how the transcriptional response of fibroblasts in HF, as captured by their component in MCP1, related to the distinct fibroblast cell-states. We hypothesized that the multicellular division of fibroblast functions during HF could be regulated in two complementary ways: first, through compositional regulation, where the population size of specific cell states adjusts to fulfill necessary functions in the tissue; and second, through molecular regulation, where global gene deregulation occurs across different cell states as a result of myocardial remodeling. To explore compositional regulation, we analyzed the genes captured in the fibroblast component of MCP1. We found that the enrichment of state-specific markers in MCP1 strongly correlated with the mean compositional changes of those states (Pearson's correlation = −0.98, $p = 10^{-16}$) (Fig. 4D). This suggests that MCP1 reflects shifts in fibroblast state composition by

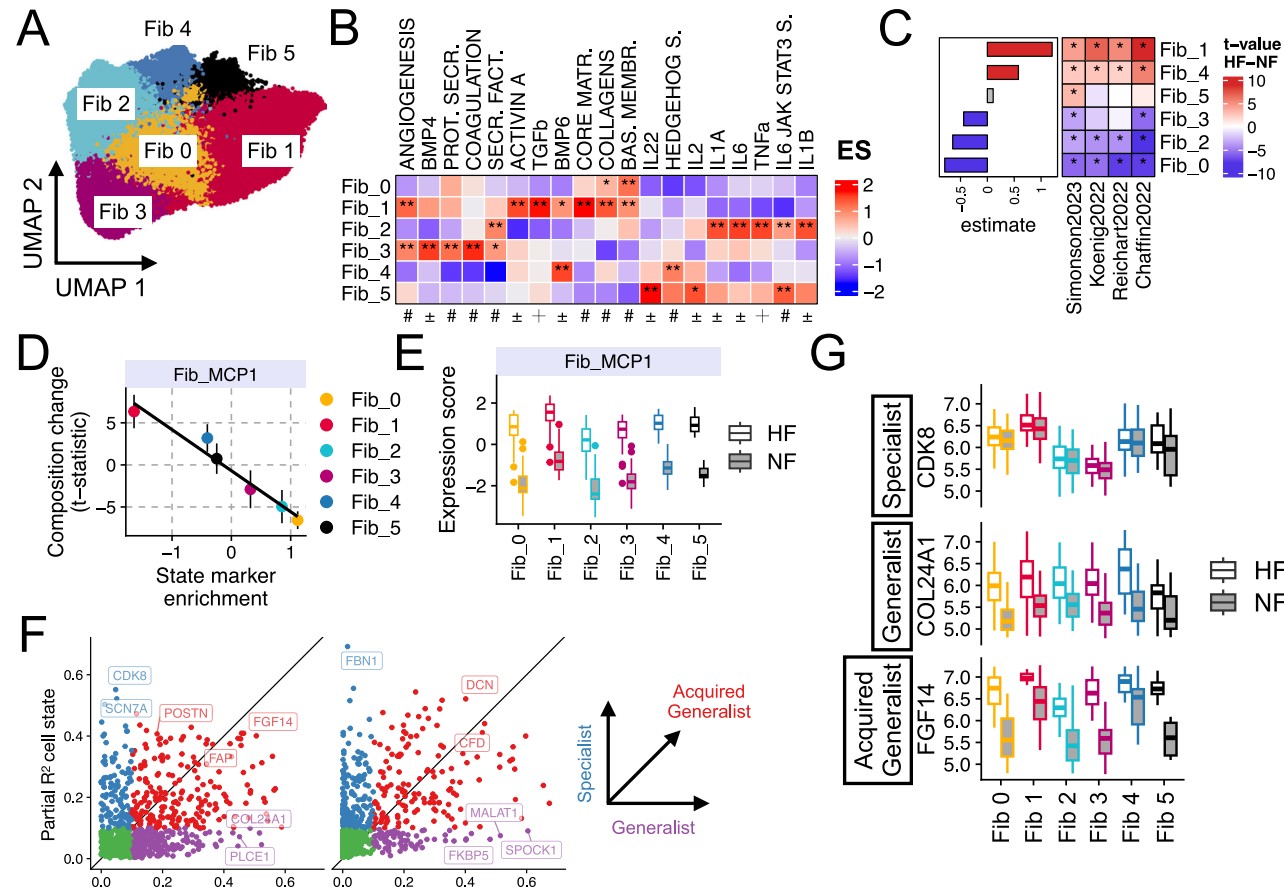

**Fig. 4 | Exploring the division of labor in fibroblast activation in heart failure.** **A** Uniform Manifold Approximation and Projection (UMAP) of 242,045 fibroblasts integrated from the four core HF studies. Cells are colored by clusters. **B** Mean enrichment scores of prior knowledge gene sets in pseudobulked states. Star indicates adj. two-sided $p$ value < 0.01 from enrichment analysis using univariate linear models. The symbols represent the database of origin: # = MSigDB, ± = Cytosig, + = PROGENy. **C** Compositional changes of fibroblast states ($y$ axis) between control and HF patients from different studies ($x$ axis) using linear models, color of tiles represents t-statistics, stars indicate adj. two-sided $p$ value < 0.05. Side bar plots display the fixed effect estimates from a linear mixed model. **D** Comparison of the compositional change of fibroblast states in HF ($y$ axis, points indicate the mean and error bars indicate standard deviation, $N = 4$ SN-core-studies) with the enrichment of the top 200 state markers in the fibroblasts component of MCP1 ($x$ axis). **E** Enrichment scores of the fibroblast component of MCP1 in pseudobulked patient profiles per fibroblast state (color) separated by failing (HF) and non-failing (NF) hearts. **F** Modeling individual gene's expression with HF and cell state covariates with linear mixed models and comparing the partial $R^2$ values for each covariate to characterize gene's expression pattern. Left and right panels display genes associated with HF and NF, respectively, according to the MCP1. **G** Normalized expression of pseudobulked patient profiles per state for selected representative markers (based in partial R2 of linear mixed models from **F**) of different divisions of labor programs: specialist (*CDK8*), acquired generalist (*FGF14*), and generalist (*COL24A1*). Sample sizes for patient profiles per fibroblast state in the HF and NF groups (displayed in **B**, **C**, **E**, and **G**) as follows: Fib_0 (HF: 51, NF: 42), Fib_1 (HF: 52, NF: 40), Fib_2 (HF: 52, NF: 42), Fib_3 (HF: 49, NF: 42), Fib_4 (HF: 40, NF: 29), Fib_5 (HF: 29, NF: 15). Boxplots in (**E**–**G**) display the minimum, Q1, median, Q3, and maximum; outliers lie beyond 1.5 times the IQR from Q1 or Q3 and are shown as points. Source data are provided as a Source Data file.

representing markers in proportion to their abundance. Next, we examined the molecular regulation of genes in MCP1 by comparing the enrichment of MCP1 genes in pseudobulk expression profiles across patient samples and cell states. While we observed variability in enrichment scores between cell states, a consistent difference emerged between failing and non-failing tissues (Fig. 4E). Linear mixed models of the enrichment scores revealed that MCP1 expression was more strongly associated with HF status (semi-partial $R^2$ 0.90) than with cell state identity (semi-partial $R^2$ 0.57). These findings characterize a phenotypic shift in fibroblasts toward an HF-specific program, as captured by MCP1. Although this shift involves an increase in the composition of Fib1 and Fib4, it is more accurately characterized by the broader acquisition of MCP1 across all fibroblast states, in line with the observations that fibroblast activation is a presumably continuous process and thus cannot be fully explained by the accretion of a state[41].

Finally, given the observed compositional and molecular regulation of fibroblast functions in HF, as captured by MCP1, we investigated

the expression variability of individual genes across different cell-states and disease statuses. We applied linear mixed models to the pseudobulk expression of fibroblast states in each patient sample, using HF status and cell-state label as fixed effects, with studies treated as random effects. Each gene was then categorized into three expression groups—specialist, generalist, or acquired generalist—based on the partial explained variance attributed to cell-state identity and HF status (Methods, Fig. 4F). This classification highlighted whether a gene's variability was driven primarily by cell-state differences (e.g., *CDK8*), HF (e.g., *COL24A1*), or both (e.g., *FGF14*) (Fig. 4G, Supplementary File 6). Notably, established matrifibrocyte (Fib1) markers, such as *POSTN*, *THBS4*, and *CILP*, were regulated in both cell-state and disease contexts, suggesting that matrifibrocyte characteristics are broadly acquired across fibroblast states (Supplementary Fig. 10F). Enrichment of the distinct expression groups in MCP1 and the HF bulk reference indicated a higher enrichment of genes with acquired generalist regulation (Supplementary Note 5, Supplementary Fig. 11G, H). Altogether, our results suggest that the expression of MCP1 fibroblast

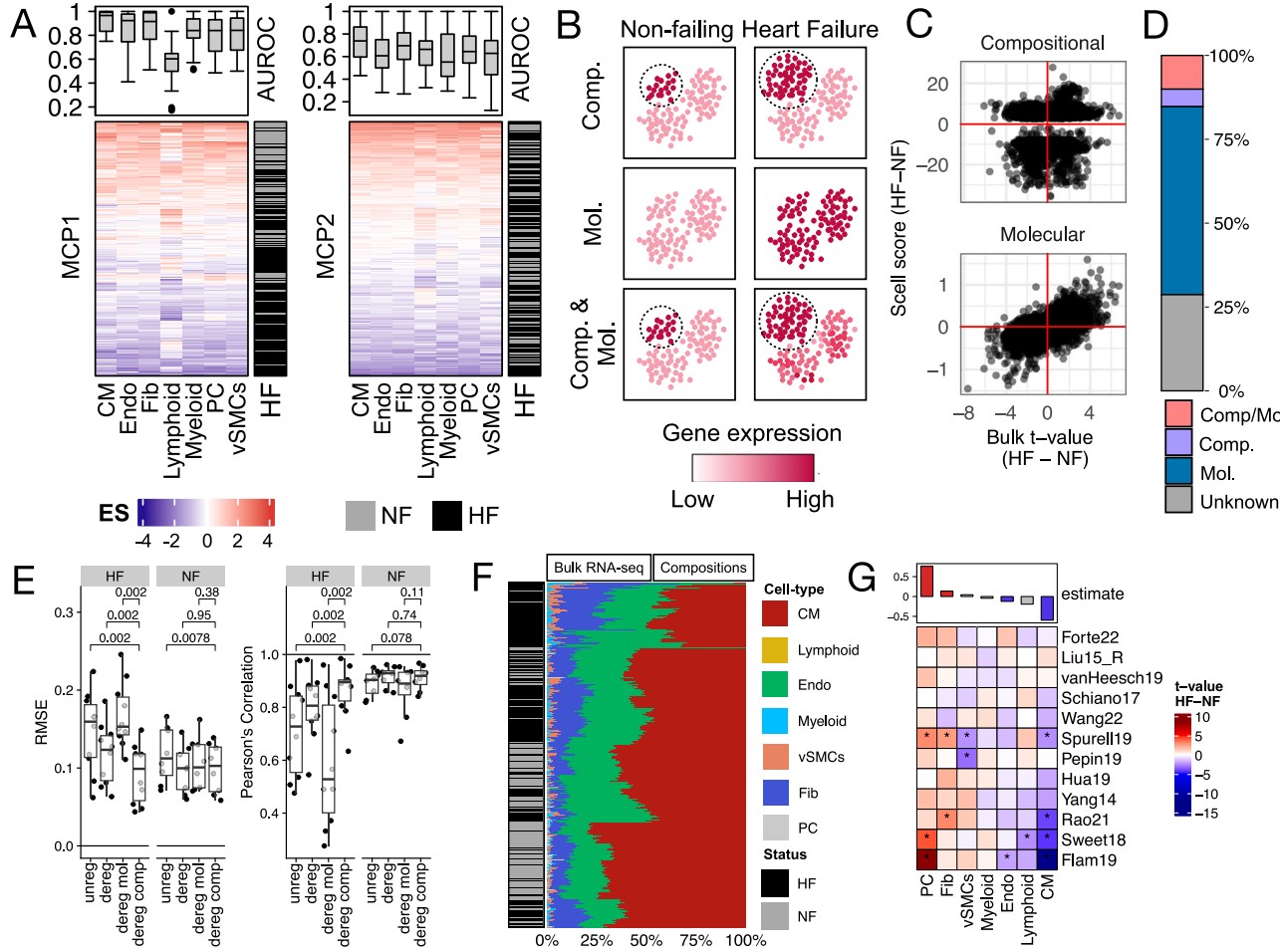

**Fig. 5 | Integrating heart failure multicellular programs (MCPs) with bulk transcriptomics. A** Cell-type specific transcriptional processes associated with MCP1 and MCP2 are enriched in every patient tissue sample from the bulk core study collection ($n = 1392$). Upper panels show AUROC distributions ($n = 21$) evaluating how well cell-type-specific responses classify non-failing hearts in each study. **B** Schematic of potential tissue-level regulatory mechanisms underlying observed gene expression changes in bulk data. Upregulation in HF may reflect increased abundance of cell types expressing the gene (Compositional regulation, Comp.), increased gene expression within cells (Molecular, Mol.), or both (Comp/ Mol). Downregulation follows similar principles. **C** Consensus gene-level statistics of HF-associated expression changes from bulk ($x$ axis) and compositional (upper) or molecular (lower) regulation from single-nucleus data ($y$ axis). Genes shown are from the top 8942 in the consensus bulk ranking (adj. Fisher $p$ value < 0.05) (Fig. 2C). **D** Annotation of deregulation events derived from the combination of bulk and single-nucleus transcriptomics studies. The annotation comes from the top 8942 genes of the consensus ranking (adj. two-sided Fisher $p$ value < 0.05).

**E** Root mean square error and Pearson correlation of pseudobulk deconvolution results (core and supporting single-nucleus data). Each dot is one dataset, stratified by tissue type (HF, $n = 10$; NF, $n = 8$) and signature gene set used ($x$ axis). Unreg, unregulated; dereg, deregulated; dereg mol, molecularly deregulated; dereg comp, compositionally deregulated. Two-sided paired Wilcoxon test. **F** Hierarchical clustering of bulk RNA-seq samples ($N = 697$) based on deconvoluted composition using a healthy reference (subset to compositionally regulated genes and seven major cell types). **G** Heatmap of $t$-statistics and estimated differences between HF and NF hearts from differential compositional analysis using $t$ tests and linear mixed models. Stars: adj. two-sided $p$ value < 0.05. In all panels cardiomyocytes (CM), fibroblasts (Fib), pericytes (PC), endothelial (Endo), vascular smooth muscle (vSMCs) cells. Heart failure (HF), and non-failing (NF) hearts. Boxplots in (**A**, **E**) display the minimum, first quartile (Q1), median, third quartile (Q3), and maximum; outliers lie beyond 1.5 times the interquartile range (IQR) from Q1 or Q3 and are shown as points. Source data are provided as a Source Data file.

component genes is distributed differently across fibroblast states, with some genes acting as specialists in non-failing fibroblasts but becoming broadly expressed across states in HF. These acquired generalist genes represent the prioritized program in both multicellular responses and bulk profiling.

**Conservation of multicellular responses in HF in bulk transcriptomics**

MCP1 was derived from 132 patients across four core single-nucleus studies and captured a shared multicellular transcriptional variation that could be functionally interpreted through a blueprint of cell dependencies, providing insights into fibroblast state coordination. However, the limited size of the patient cohorts could limit their generalizability to a broader patient population. To address this, we

investigated whether the MCPs could be detected in our compiled larger bulk HF cohort -10 times the size of the SN collection. We enriched bulk transcriptomics samples from each study in our core collection with MCP1 and MCP2 cell-type signatures and compared their ability to distinguish failing from non-failing patients (Fig. 5A). MCP1 showed stronger performance, with a median AUROC of 0.84 and a median silhouette score of 0.4 across patient groups, while MCP2 had a median AUROC of 0.64 and a median silhouette score of 0.04, indicating broader applicability of MCP1 across studies and technologies.

Based on these results, we hypothesized that the molecular information captured by the core HF bulk data reflects both cell-type composition changes and multicellular gene expression responses of MCP1 within the tissue (Supplementary File 7). Specifically, changes in

gene expression from bulk data in HF could be driven by shifts in the abundance of cell types that express the gene (compositional regulation), transcriptional regulation that occurs independently of cell-type composition (molecular regulation, either within a single cell type or across multiple cell types), or a combination of both mechanisms. To identify which process drives the consensus transcriptional bulk signature of HF, we correlated the consensus effect size of differentially expressed genes in bulk data with compositional and molecular deregulation scores derived from core single-cell studies (Methods). For the top 8942 genes in the bulk consensus signature (adj. Fisher $p$ value < 0.05), we observed a Spearman correlation of 0.14 with compositional scores and 0.6 with molecular scores, indicating that multicellular transcriptional coordination, captured by MCP1, is the dominant factor influencing gene expression changes in HF bulk transcriptomics (Fig. 5C).

We annotated each of the top 8942 genes in the bulk consensus signature (adj. Fisher $p$ value < 0.05) as potentially deregulated by compositional, molecular, or a combination of both mechanisms. This classification was based on the agreement between their differential expression size-effect in bulk and the molecular and compositional scores derived from single-nucleus data (Fig. 5C). For 29% of the genes, we were unable to assign a mechanism due to incongruences in the direction of regulation or limitations in single-nucleus gene coverage. However, 5% of the genes were linked to compositional regulation, 56% to molecular regulation, and 10% to both, underscoring that bulk transcriptional changes in HF cannot be reduced to shifts in tissue composition alone (Fig. 5D).

### Cell type composition estimation in bulk transcriptomics

The multicellular transcriptional responses during HF captured by MCP1 were consistently observed in patient cohorts profiled using both single-nucleus and bulk RNA-seq studies. However, it remained unclear whether the changes in cell type composition observed in single-nucleus studies during HF could be generalized to larger bulk cohorts. To address this, we estimated cell-type compositions from bulk expression data in the core bulk collection using deconvolution methods. These methods infer cell-type proportions by learning the relationship between cell-type marker expression and cell-type abundance from single-cell data. To achieve reliable performance in HF data, it is crucial to properly define cell-type markers and account for their expression changes in HF.

We first noted that 33–47% of cell-type markers identified from our core single-nucleus studies (i.e., genes whose expression is characteristic of a cell type) were deregulated in HF (Supplementary Fig. 12A, B). In the previous section, we found that many deregulated genes could not be explained solely by changes in cell type composition. For example, *NPPA*, a cardiomyocyte marker, is upregulated in failing heart tissue despite a decrease in cardiomyocyte abundance, indicating that *NPPA* is regulated more by molecular mechanisms than compositional ones. We hypothesized that molecularly regulated cell-type markers like *NPPA* could reduce deconvolution accuracy, as these markers would not reliably reflect true cell type composition. To test this, we evaluated different sets of cell-type markers in estimating cell-type compositions from ten single-cell studies aggregated into pseudobulks with known compositions (Supplementary Fig. 12D, E; Supplementary Note 6). Our benchmarks confirmed that molecularly regulated markers performed poorly as indicators of cell-type composition in disease. Moreover, using only compositionally regulated markers significantly improved deconvolution results in HF samples (Fig. 5E).

Using compositionally regulated cell-type markers (Supplementary File 8), we estimated cell-type compositions across all 697 patient tissues from the subset of 12 RNA-seq studies in the core bulk collection. We found that cardiomyocytes followed by endothelial cells and fibroblasts, were the most abundant cell types (Fig. 5F). However, cell-

type compositions only modestly grouped failing and non-failing patients (median silhouette width HF = 0.19, NF = 0.4, one-sample $t$ test, $p$ values: HF < $10^{-8}$, NF < $10^{-39}$) (Supplementary Fig. 12F). Differential compositional analysis between failing and non-failing patients showed little agreement on cell-type changes between studies (Fig. 5G). There was a modest trend showing a decrease in cardiomyocytes and endothelial cells, and an increase in pericytes and fibroblasts (linear mixed model adjusted $p$ value < 0.05). However, variability between studies was high, particularly in cardiomyocytes, fibroblasts, and endothelial cells (Supplementary Fig. 12G).

These findings did not align well with the composition changes observed in single-cell data, except for a shared decrease in cardiomyocytes. When comparing single-cell and bulk compositions, we found that 80% of the variance was associated with data modality and study labels, while only 53% was associated with HF status based on principal component (PC) analysis (Supplementary Fig. 12H). In summary, the agreement between bulk studies on cell type composition changes was modest, and the compositional changes did not match those from single-nucleus studies. These discrepancies might be due to technical factors, such as cell isolation protocols or errors in bulk deconvolution. On the other hand, we observed a strong agreement between MCP1 and the bulk cohort. This difference between molecular and compositional changes suggests that the observed molecular shift is primarily driven by shared multicellular coordination within the tissue, independent of cell type composition.

### Reinterpreting independent data with the MCPs of HF

Finally, to illustrate the potential of the robust MCPs in contextualizing and integrating independent datasets, we projected supporting single-nucleus and single-cell studies onto our patient map (Fig. 6A, B). We confirmed the generalizability of the HF processes described by MCP1 and MCP2, with perfect classification of 12 tissue samples from HF patients across hypertrophic cardiomyopathy (HCM), ICM, and cardiac sarcoidosis etiologies, drawn from two studies without healthy reference patients (Fig. 6B). Projection of myocardial infarction and congenital heart disease patients from two additional studies highlighted the activation of apoptotic processes related to MCP2 (Supplementary Fig. 13A, B, Supplementary Note 7). Additionally, we observed little agreement between the multicellular responses in mouse models of heart disease and those captured by the core SN HF studies (Supplementary Fig. 13C, D, Supplementary Note 7). By projecting these supporting datasets onto the multicellular patient map of HF, we aimed to describe an analysis strategy that allows for the comparison of HF studies across diverse etiologies, biological contexts, and species.

We used two supporting studies to examine myocardial remodeling and its reversal, specifically the reactivation of fetal programs and cardiac recovery following LVAD implantation. Tissue samples from fetal hearts, pediatric dilated cardiomyopathy patients, and non-failing hearts showed differences in MCP1 activation consistent with HF progression (ANOVA, $p$ value = 0.04), with the strongest difference observed between fetal and non-failing hearts ($t$ test, adj. $p$ value = 0.047; Fig. 6C). These findings link the generalist MCPs of HF to the reactivation of fetal myocardial programs, a hallmark of the disease. Direct comparison of gene activation in fetal versus failing hearts revealed shared patterns in cardiomyocytes, including reduced activity of PPAR-α and PPAR-δ footprints and decreased fatty acid oxidation, alongside increased BMP6 footprints. In fibroblasts, we observed increased Hedgehog signaling, TGFB1, IL4, and GLI3 footprints, and enhanced collagen fibril organization (Fig. 6D), providing detailed insights into fetal reprogramming captured by MCP1. Heart tissue samples from recovered and non-recovered DCM patients, before and after LVAD implantation, also aligned with MCP1 activation (Fig. 6E). Among recovered patients, we observed a significant change in MCP1 activation after LVAD implantation, where

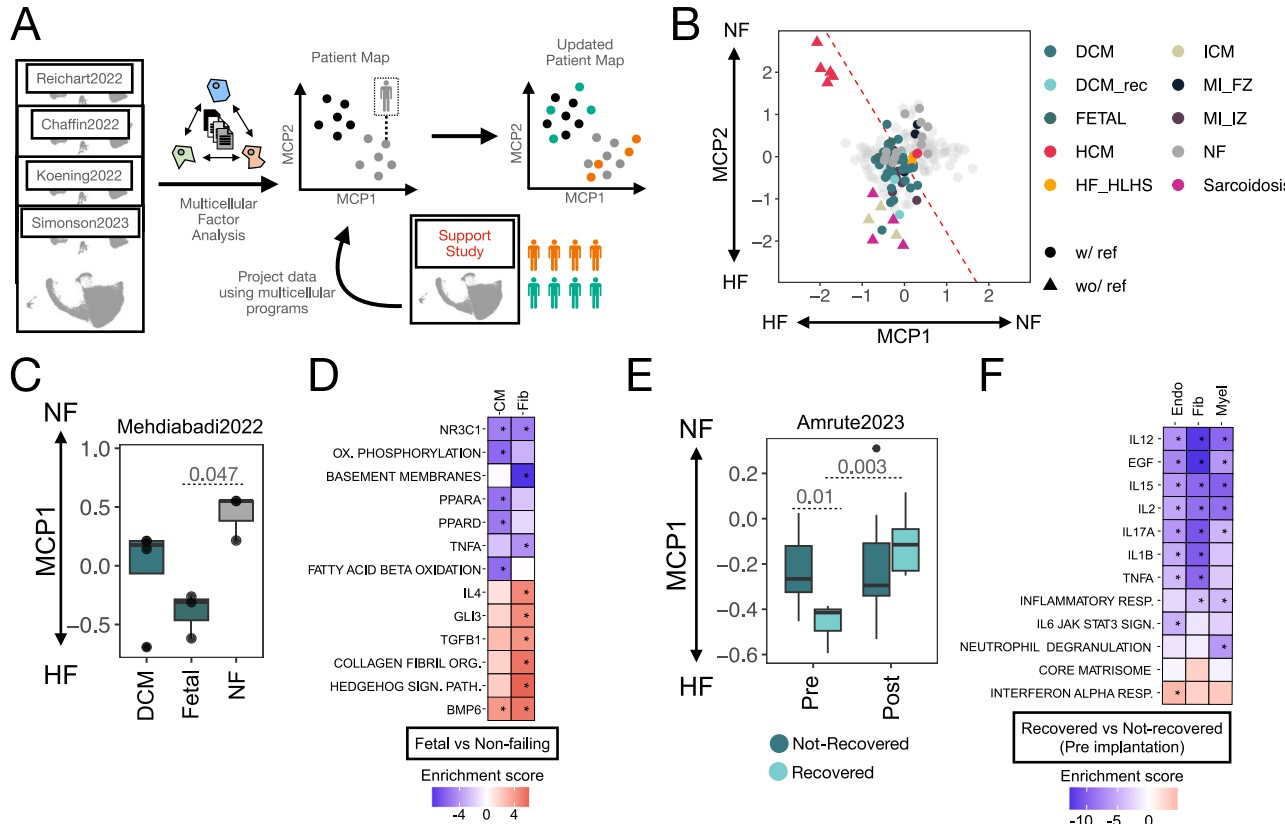

**Fig. 6 | Reinterpretation of the patient map using supporting external studies.**
**A** The patient map built with the single-nucleus (SN) core studies can be used for the reanalysis of datasets of similar contexts. One possibility is to project the tissue samples of an independent dataset into the latent space defined by multicellular programs (MCPs) describing the variability of tissue samples from the SN core studies. The values of activation of the different MCPs can then be evaluated for tissue samples of the projected studies. **B** Tissue samples of supporting studies projected into the patient map of HF built from the SN-core-studies (light gray). Color of the points denotes distinct etiologies, and shape of the points if the study did or did not contain or reference control samples. Dotted line represents a decision boundary for the classification of HF samples. coming from a logistic regression model fitted in the SN-core-study collection (see Methods). **C** MCP1 values of projected heart tissue samples of dilated cardiomyopathy pediatric patients ($n = 4$), fetal hearts ($n = 3$), and non-failing controls ($n = 3$). Adjusted two-

sided $p$ value of a $t$ test comparing the mean MCP1 scores between fetal and non-failing samples. **D** Enrichment scores of differentially active or inactive processes in fetal hearts over non-failing donor hearts captured by MCP1. Univariate linear models with adjusted two-sided $p$ value. **E** MCP1 values of projected paired HF samples obtained from dilated cardiomyopathy patients before (*pre*) and after (*post*) implantation of a left ventricular assist device. Colors denote the response to treatment (recovered = 10, not recovered = 16). Adjusted two-sided $p$ values come from a $t$ test of *pre* samples, and a paired $t$ test of recovered samples *pre* and *post* implantation. **F** Enrichment scores of differentially active or inactive processes in recovered patients versus not recovered captured by MCP1. Univariate linear models with adjusted two-sided $p$ value. Boxplots in (**C**, **E**) display the minimum, Q1, median, Q3, and maximum; outliers lie beyond 1.5 times the IQR from Q1 or Q3 and are shown as points. Source data are provided as a Source Data file.

post-implantation samples resembled those from non-failing hearts (paired t-test, adj. $p$ value = 0.003). Pre-implantation samples also showed differences in MCP1 activation between recovered and non-recovered patients ($t$ test, adj. $p$ value = 0.012), indicating that this MCPs reflects disease severity and can offer insights into recovery potential. Comparison of gene expression between recovered and non-recovered patients before implantation revealed alignment with MCP1 primarily in endothelial cells, fibroblasts, and myeloid cells. In all three cell types, we observed coordinated downregulation of inflammatory responses, including cytokines (IL17a, IL2, IL15, TNFα) and growth factors (EGF) (Fig. 6F). These results suggest that lower inflammation levels may be linked to improved recovery in HF patients post-LVAD. We evaluated whether clinical features were associated with MCP1 activation levels and the expression of related genes between recovered and non-recovered patients. No significant associations were found, suggesting that MCP1 captures treatment response-related information that is independent of the measured clinical variables.

In conclusion, the robust MCPs captured in this study, particularly MCP1, provide a powerful framework for understanding the molecular mechanisms underlying HF, with broad applicability across various HF

etiologies, biological contexts, and species. By enabling the integration and comparison of independent studies, this multicellular reference map offers a valuable tool for clinical applications, such as identifying patients who may benefit most from targeted interventions, predicting disease trajectories, or evaluating animal models of disease.

## Discussion
The molecular processes driving HF have been extensively studied through large-scale transcriptomics, both at bulk and single-cell resolution. However, these studies often present a fragmented and sometimes inconsistent view of transcriptional regulation, with variability arising from differences in patient cohorts and technologies. To fully benefit from these molecular profiling efforts and create a meaningful clinical impact, an integrative approach is needed to identify robust, consistent disease processes in HF. This integration must overcome two main challenges: first, the harmonization of diverse data acquisition protocols, storage formats, and accessibility to ensure effective use by both computational and clinical scientists; and second, the use of statistical methods that account for the multicellular nature of tissues, enabling joint analysis of bulk and single-cell data while supporting the integration of new datasets and

technologies. In this study, we addressed these challenges by combining single-nucleus transcriptomics with bulk transcriptomics from larger patient cohorts to build a consensus of multicellular gene expression changes during cardiac left ventricular remodeling. This expanded upon our previous reference of the HF transcriptome[16] and uncovered reproducible molecular insights from multiple independent studies. Rather than focusing solely on cell-type taxonomies[44], as is common in single-cell analyses, our work captures MCPs that describe coordinated gene expression across multiple cell types. This approach shifts attention to patient-level variability and tissue-wide molecular coordination, offering new insights into how HF affects the heart as an integrated system.

Our comprehensive comparison of single-nucleus and bulk transcriptomics datasets revealed a strong consistency in gene expression responses during HF, despite variations in technical and clinical factors across studies. The addition of new bulk transcriptomics data allowed us to refine the ranking of genes most closely associated with HF, reinforcing our previous findings that coordinated molecular responses during end-stage HF are conserved across patient cohorts[16]. By analyzing MCPs derived from single-nucleus data, we were able to break down these molecular responses into cell-type-specific programs, which were also detectable in bulk transcriptomics, further confirming the robustness of these convergent molecular responses.

Inference of cellular dependencies from MCPs provided new insights into previously unexplored aspects of gene regulation in HF. Analysis of MCP1, the largest axis of conserved variation, showed that cells in both failing and non-failing hearts follow similar coordination patterns across cell types. Fibroblasts emerged as central to MCP1, having the strongest influence on the gene expression of other cell types, particularly cardiomyocytes. These dependencies suggest that when fibroblasts express their specific MCP1 program, other cell types, such as cardiomyocytes, respond by expressing their corresponding set of genes. However, we observed a weak correlation between ligand-receptor coexpression and gene expression coordination, indicating that cellular coordination in HF may extend beyond direct cell-to-cell communication through ligand-receptor signaling[45-47] and reflecting the limitations of studying ligand-receptor interactions with transcriptomics[48]. Further investigation into cardiomyocyte-fibroblast dependencies identified ligand-receptor pairs associated with ECM components, reflecting how ECM changes can drive transcriptional, morphological, and functional shifts in cardiomyocytes during HF[33,49]. In addition, we found that these ligands could represent a fraction of potential inducers of the stress response of cardiomyocytes, traced in the expression of genes such as *ANKRD1* or *CCND1*. Our findings suggest that this multicellular representation of molecular processes in HF provides a valuable framework for studying disease at the tissue-level and could guide the identification of therapeutic targets that stabilize multicellular coordination and maintain tissue homeostasis.

Multicellularity allows tissues to perform functions that individual cells cannot achieve alone, with a key strategy being the division of labor, where tasks are distributed among cells. This can range from highly specialized cells performing specific tasks to generalist cells covering a broad range of functions. Previous studies have shown that tissue function typically arises from a balance of specialist and generalist cells, which can dynamically shift roles depending on the context, such as in disease[50-53]. By comparing gene expression across cell types within MCP1, we found that most dysregulated genes were cell-type specific, indicating that cell lineages act as specialists. However, when analyzing fibroblast function in greater detail, this pattern shifted. MCP1 showed a broad activation across all fibroblast states. Upon further decomposition of this Fib component of MCP1 into generalist or specialist expression patterns, we found that matrifibrocyte markers formed an acquired generalist program, as all fibroblast states upregulated these markers in HF. This suggests that generalist gene programs, rather than highly specialized ones, drive fibroblast

adaptation to HF, which was confirmed by their high ranking in the consensus bulk signature. Additionally, we identified a purely generalist program that included less studied genes, such as *COL24A1*, previously linked to osteoblast differentiation[54], but with an unknown role in cardiac function. While fibroblasts are expected to reallocate resources to meet tissue demands, different regulatory strategies likely drive the expression of these gene groups through distinct mechanisms. These findings improve our understanding of tissue dynamics in cardiac fibrosis and may help identify biomarkers for diagnostic or therapeutic use by prioritizing robust markers that generalize to the broader HF population. However, this analysis is limited by the resolution of the defined cell states and the integration strategy used, and further work at the single-cell level, along with experimental validation, is needed to fully explore these regulatory patterns.

The integration of bulk and single-nucleus data, in addition, opened the opportunity to study potential tissue-level mechanisms of deregulation of genes consistently associated with HF. Particularly, we were interested in understanding whether the source of gene deregulation in HF was linked to changes in tissue composition (i.e., an increase or decrease in specific cell types) or molecular mechanisms independent of cell-type composition. We found that conserved disease programs primarily operated independently of tissue composition, suggesting that the multicellular coordination events during cardiac remodeling represent a broader disease mechanism affecting the heart at a systemic level. For example, the cardiomyocyte marker *NPPA*, a recognized HF biomarker, showed increased expression despite a reduction in the number of cardiomyocytes. This highlights how gene expression changes can occur through mechanisms beyond shifts in cell type abundance. Moreover, identifying the source of expression changes not only provides biological insights but also improves the accuracy of computational methods for estimating cell-type proportions in HF from bulk transcriptomics. Our ability to trace multicellular disease processes inferred from single-nucleus data in bulk transcriptomics supports the notion that these molecular processes can occur independently of local tissue composition. This is further evidenced by the convergence of molecular signals across tissue samples collected from distinct locations in the left ventricle of different patients. However, it is important to acknowledge that cell-type compositions in these analyses were approximated from bulk and single-nucleus data, and may be subject to biases.

The integration of transcriptomics data across scales enabled us to build a patient map that captures heart tissue variability across cell types, based on the activation of two distinct MCPs (MCP1 and MCP2). Interpreting this variability with clinical covariates could help identify molecular markers of heart disease. We showed that MCP1 and MCP2 successfully distinguish failing from non-failing hearts across core studies. Although the patient map did not show strong associations with expected clinical covariates such as age, sex, or etiology, our analyses suggest that this reflects limited variability in these covariates within the collected patient cohorts rather than a limitation of the modeling strategy. Despite this, we propose that the patient map has translational potential as a reference for analyzing single-cell data from related HF contexts. By projecting independent datasets onto MCP1 and MCP2, we facilitated the reinterpretation of studies on fetal samples, myocardial infarction, animal models, and LVAD-treated patients. For example, MCP1 separated fetal from control samples by identifying shared deregulation of genes, such as BMP6 footprints in cardiomyocytes and hedgehog signaling in fibroblasts. Early cardiac remodeling after infarction, however, aligned more with MCP2. Our findings also indicated a lack of concordance between the multicellular disease processes of HF in human tissue with the ones observed in commonly used mice models (TAC, Angiotensin II). This could be linked to the difficulties in determining disease time-points in mice that are aligned with end-stage HF patients, the lack of large animal cohorts, the larger

amount of tissue profiled in mice compared to human tissue or species differences in pathway activation. Importantly, the decision to implant an LVAD is complex and could be facilitated from more accurate predictions of recovery success. While studies have used clinical data to predict risk and benefit, molecular data provides another layer of insight. While the patient cohort in the study by Amrute et al [6]. was clinically indistinguishable, their recovery trajectory was captured by a regression on MCP1, suggesting a molecular profile that could predict recovery pre-implant. These findings indicate that the molecular state of a patient, as described by her position within the HF patient map, may provide clinically meaningful information beyond conventional clinical metrics like LVEF or demographic factors. In this regard, molecular phenogrouping studies have indicated that molecular data can correlate with distinct clinical subgroups[55,56], an approach of great interest for personalizing treatment for HF patients. Our study aimed at identifying and quantifying a shared signal, thus this molecular heterogeneity between individuals could be disregarded as statistical noise. We propose that these approaches, capturing both consensus and heterogeneity, are not mutually exclusive. Rather, the consensus derived from our study is more likely to generalize to the HF patient population and can enhance precision medicine approaches, by capturing HF patient heterogeneity along the identified axes of MCPs as demonstrated by the molecular reinterpretation of myocardial infarction, fetal reprogramming, and LVAD response.

Through our curation efforts, we also gathered insights into the clinical spectrum of HF patients studied over recent decades, revealing a predominance of white males with end-stage disease. This demographic bias may limit the generalizability of findings to a broader HF population—particularly patients of diverse geographic origins, female sex, those with HF with preserved ejection fraction, early-stage HF, or HF arising from etiologies such as inflammatory or infiltrative cardiomyopathies. Additionally, our work highlighted the critical need for sharing comprehensive metadata to support clinically relevant reinterpretation of the data. Addressing these two challenges is essential for advancing inclusivity and clinical applicability in the field.

Overall, our study provides insights into the conserved mechanisms of HF remodeling and multicellular organization supported by the largest collection of heart-specific multi-scale transcriptomics to date. We anticipate that this resource will serve as a valuable reference for understanding the coordinated transcriptomic changes in end-stage HF. To facilitate its use by both computational and clinical scientists, we have made this resource available in various formats through an accessible platform (https://saezlab.shinyapps.io/reheat2/, Supplementary Note 8, Supplementary Fig. 14), enabling data-driven translational research.

## Methods

### Study the inclusion criteria of single-nucleus studies

We collected human HF single-nucleus transcriptomic studies reported in the literature with available gene expression count matrices in public repositories. We included studies whose experimental design consisted of profiling biopsies of human heart samples of the left ventricle with free wall or apex sublocations from patients with end-stage HF and patients with non-failing hearts. In addition, we prioritized studies that profiled at least 10 patients per group. Protocol type and links to source data are presented in Table 1 and Supplementary File 1. We collected available metadata from data repositories and published manuscript files from the referenced studies. Race information is reported as collected from the original study authors.

Single-nucleus studies that did not fit the inclusion criteria because of the lack of independent profiling of patients with non-failing hearts, acute HF, and pediatric or fetal samples were analyzed independently. In addition, two studies profiling established models of HF in mice (Angiotensin II-Induced (AngII) and transverse aortic constriction (TAC)) were curated to evaluate their consistency with human

data. Information on these supporting studies is provided in Table 1 and Supplementary File 1.

### Processing of single-nucleus studies

Expression count matrices of the selected studies were downloaded directly from their specific repository links. Studies originally provided as R objects were transformed into anndata objects using Zellkonverter. Studies providing count matrices with Ensembl IDs (Reichart 2022, Kuppe 2022) were transformed into gene symbols using biomaRt v2.58.0[57] and summing all reads of Ensembl IDs assigned to the same symbol. Across studies, we filtered out samples belonging to right ventricles if available, and the unit of each study was considered to be the patient in case multiple samples were collected, as in Reichart (2022). No further processing was performed on studies, since the original authors provided processed data. To ensure the comparability of the analysis across atlases, we defined a heart cell ontology that included the following cell types: CMs, Fibs, Endos, PCs, vSMCs, and myeloid and lymphoid cells. Regular expressions over original cell annotations provided by the selected studies were performed to align each dataset to our proposed ontology. Unannotated cells were discarded. Single-nucleus studies were transformed into collections of pseudobulk expression profiles by summing up the counts of all cells belonging to each of the cell types defined in our ontology for each sample. Quality metrics, including the number of cells used, the total number of reads, and the number of genes with available counts, were calculated for each pseudobulk sample. To estimate the agreement between the new assigned cell annotations, we compared the overlap of marker genes of the seven cell types estimated from each study independently using Jaccard Indexes. Marker genes of each cell type per study were calculated using differential expression analysis between the pseudobulk expression profiles across samples of one cell-type versus the rest using edgeR v4.0.2 with default parameters of gene filtering. Genes with a log-fold-change over 2 and a false discovery rate (FDR) lower than 0.01 were considered cell-type markers.

To estimate the levels of noise expression values within the pseudobulk expression profiles across studies, we defined a score that quantified the amount of contamination either by the misannotation of single cells or by background expression. Our contamination score was defined for each pseudobulk expression profile as the ratio of the number of reads belonging to contaminating genes to those belonging to marker genes. Given a pseudobulk expression profile belonging to a cell type, we defined as contaminating genes all the marker genes of the rest of the cell types. In case a marker gene was expressed in more than a single cell type, the reads of the gene were always assigned to the marker gene set of the cell type tested. The gene set of marker and contaminating genes was defined for each study independently.

### Processing of supporting studies

The seven supporting studies (Table 1) were processed depending on available data formats. For studies where only cell ranger output files were available (Liu2022, Mehdiabadi2022, McLellan2020), we used a uniform processing pipeline with the Seurat v5.0.3 R-package[58]. This included a filtering procedure per sample, doublet detection with DoubletFinder[59], cell filtering (mitochondrial percentage <0.1 in single-nucleus and <0.2 in single cell, genes per cell >300, RNA counts per cell >500, ribosomal genes per cell <0.1) in samples with at least 10 cells and 250 features. Samples were integrated on joint highly variable genes with Harmony v1.2.0 R-package[60]. Leiden clustering was applied to identify cell lineages, and overrepresentation analysis of consensus cell type markers from core HF studies was used to annotate clusters; cell clusters that did not match the seven cell lineages were discarded. For studies with available count matrices and cell lineage annotations (Hill2022, Nicin2022, Ren2020, Litviňuková2020, Kuppe2022), we mapped annotations to our harmonized vocabulary. All studies were

then summed to pseudobulk profiles and normalized via edgeR v3.36.0.

## Processing and analysis of core HF bulk studies

For the addition of the five bulk studies, we applied a similar strategy as reported previously[16]. In brief, we identified five studies that fulfilled the inclusion criteria for bulk studies. We downloaded sequencing files in FASTQ format from Wang22 and Forte22 and realigned data to the human genome as implemented in the ArchS4 pipeline in biojupies[61]. Raw counts were downloaded for Flam19, Rao21, Hua19 from repositories (Supplementary File 1). For all studies meta metadata was aligned to a shared vocabulary. Low-expressed genes were filtered, and counts were normalized with TMM (edgeR v3.36.0), log-scaled, and voom-transformed (limma v3.50.3). Differential gene expression analysis, study comparison, classification, and gene expression meta-analysis were performed as described[16]. Briefly, we assessed the ability of individual studies to classify samples from other studies by defining a disease score. The disease score linearly combines a sample's gene expression values with a disease pattern from an independent reference study, captured by $t$ values from differential expression analysis. This estimates how closely a sample's expression profile aligns with the reference phenotype, emphasizing coordinated gene regulation over mean expression changes. To assess whether classification performance reflected consistent transcriptional regulation, we split the top 500 differentially expressed genes (sorted by $p$ value) into up- and downregulated sets and tested their enrichment in other studies' gene-level statistics via GSEA, with genes ranked by $t$ value. The consensus signature was derived by combining the Benjamin–Hochberg corrected $p$ values of the differential expression analysis for all genes that were measured in at least 10 data sets using a Fisher's combined probability test. The updated consensus was compared to the previous ranking by Jaccard indices. For a given ranking segment, genes were selected from both rankings and tested for their classification performance and their enrichment scores across studies[16].

## Definition of consensus cell-type gene markers from single-cell data

For each cell type, we combined the FDR of the differential expression analysis for all genes that were measured in at least three studies using Fisher's combined probability test with survcomp v1.5[62]. The degrees of freedom for the significance test of each gene were defined by the number of data sets that included it. A ranking of marker genes was generated by the corrected p-value of the combined test and the mean log fold change across studies. Correction was performed with the Benjamini–Hochberg (BH) procedure.

## Compositional data analysis of core single-cell studies

Unsupervised analysis of cell type compositions was done using hierarchical clustering over the Euclidean distance matrix of all samples across studies. General and study-specific HF compositional signatures were generated using differential compositional analysis. First, cell-type compositions of each sample were transformed into centered-log-ratios (clr) using *compositions v2.0-8*[63]. Then, for each study and cell-type, a $t$ test comparing the means of clr values of failing and non-failing patient samples was performed. To test the ability of study-specific compositional disease signatures to classify samples of other studies, we defined a compositional disease score. Following our previously defined disease score, we linearly combined the scaled clr values of the samples of one study with the compositional disease signature of the rest, captured by the t-values of their compositional differential analysis. Areas under the AUROC were used to test the classification accuracy, where the non-failing class was defined as the response variable, and to measure the conservation of HF compositional changes. General HF compositional signatures across studies were estimated by modeling the difference between the means of clr

values between failing and non-failing hearts using linear mixed models. For each cell type, a model was fitted assuming random effects from studies. Across tests, $p$ values were corrected using the BH procedure.

## Multicellular factor analysis of individual studies

We summarized the molecular variability across cell types and patient samples for each study in terms of MCPs. MCPs are latent variables that capture a certain fraction of gene expression variability across samples and can be calculated from the collection of pseudobulk expression matrices of each cell type for each study. For a given MCP, each gene of each cell type gets assigned a weight that represents its importance in defining the MCP. At the same time, the activation values of an MCP across samples define a range of molecular phenotypic variability that can be associated with patient annotations to facilitate interpretability. Moreover, MCPs capture fractions of the total variance of the dataset, and these are specific for each cell type. Finally, the latent space formed by MCPs defines a patient map where new data can be projected (Supplementary Fig. 5B).

We estimated 10 MCPs for each individual study with Multicellular Factor Analysis[22] using MOFA2 v1.12[64,65]. In each factor analysis model, we included only pseudobulk expression profiles calculated from at least 20 cells and cell types profiled in at least 40% of the patient samples with more than 50 genes. Genes with less than a minimum of 20 counts in a single sample or detected in less than 40% of the remaining samples were discarded. Each filtered pseudobulk matrix was normalized using the trimmed-mean of M values method in edgeR v4.0.2 with a scale factor of 1 million and log-transformed. In each pseudobulk expression matrix of each cell type, we filtered out marker genes of each other cell type to reduce the levels of contamination. Finally, we filtered out samples within each pseudobulk expression matrix of each cell type with less than 97% of the genes measured to avoid MCPs related purely to coverage. Feature-wise sparsity was not included in the model to obtain a greater number of genes associated with MCPs. We associated the MCPs values with clinical covariates using Analysis of Variance (ANOVAs) or linear models for categorical and continuous covariates, respectively (corrected $p$ value lower than or equal to 0.05). Across tests, $p$ values were corrected using the BH procedure. To quantify the total amount of explained variance ($R^2$) associated with a clinical covariate, we summed the $R^2$ of each MCP associated with that covariate.

## Comparison of MCPs across studies

To compare the conservation of multicellular responses associated with HF across the different core single-cell studies, we extended our disease score strategy to MCPs. Similarly, as our bulk and compositional disease score, the idea behind this strategy is to show that disease signatures of one study are sufficient to classify HF samples from any other study. We interpreted the overall classification performance of the disease signatures as their level of generalizability. First, for each study, we trained a classifier of failing and non-failing hearts using the MCPs' scores of each patient and linear discriminant analysis (LDA). Then, to test the performance of each study to classify patient samples from the rest of the studies, we projected the samples of the rest of the studies into the MCPs' latent space and then predicted their disease status with the LDA classifier. In detail, to project data into an MCP latent space, we multiplied the Moore-Penrose generalized inverse matrix[66] of the concatenated gene weights across MCPs of the reference study with the scaled and normalized gene expression data of the target study. Normalization was performed following the same standards as those used to run multicellular factor analysis. MASS v7.3-57 function ginv() was used to calculate the inverse matrix of the feature weights. AUROCs were used to test the classification accuracy as mentioned previously. To evaluate the agreement between MCPs at the cell-type level, we constrained pairwise study projections to use

only the gene weight across MCPs to contain only the information of one cell-type at a time. MCPs latent space was visualized in two dimensions using Uniform Manifold Approximation and Projection (UMAP).

## Consensus MCPs estimation across studies

To estimate the consensus MCPs describing the variability of patient samples across studies, we fitted a multicellular factor analysis model to the joint collection of pseudobulk normalized expression profiles of cell types across studies. Gene expression processing was identical to the one performed for the models of individual studies, with the difference being the definition of background marker genes. Marker genes of each cell type were obtained from the consensus marker genes obtained from the combined test as previously described, with an adjusted $p$ value less than or equal to 0.0001 and a mean log fold change greater than 2. A MOFA joint model using an extended group-wise prior hierarchy was used to integrate all studies in the inference of the MCPs. Associations of the MCPs' scores with patient covariates were done with ANOVAs and linear models as performed in individual studies. To evaluate the association of MCPs' scores with clinical features in HF patients, we used linear mixed models to model MCP scores with each clinical covariate independently and using the studies as random effects. P-values were adjusted using the BH procedure.

## Supervised analysis of gene expression of HF patients

To complement our unsupervised multicellular factor analysis, we examined associations between gene expression variability in HF patients and sex, age, and etiology. For sex and age, we fitted linear mixed models to all samples, incorporating an interaction term between HF and either sex or age, with studies treated as random effects. To assess differences across etiologies, we applied a linear mixed model to HF samples, using etiology as a fixed effect and study as a random effect. Due to sample size constraints, only DCM, HCM, and ICM samples were included. DCM samples were used as the reference intersect in the model. Additionally, gene expression in HF samples was centered and standardized relative to non-failing samples $((\text{expression}-\text{mean}^{NF})/\text{SD}^{NF})$ with centering performed independently within each study. $P$ values were adjusted for multiple testing using the BH procedure.

To complement our unsupervised multicellular factor analysis, we examined associations between gene expression variability in HF patients and sex, age, and etiology. For sex and age, we fitted linear mixed models to all samples, incorporating an interaction term between HF and either sex or age, with studies treated as random effects. To assess differences across etiologies, we applied a linear mixed model to HF samples, using etiology as a fixed effect and study as a random effect. Due to sample size constraints, only DCM, HCM, and ICM samples were included. DCM samples were used as the reference intersect in the model. Additionally, gene expression in HF samples was centered and standardized relative to non-failing samples $((\text{expression}-\text{mean}^{NF})/\text{SD}^{NF})$ with centering performed independently within each study. $P$ values were adjusted for multiple testing using the BH procedure.

## Functional interpretation of MCP1 and MCP2

To facilitate the interpretation of the MCPs we curated a collection of gene sets from MSigDB[67] representing molecular and cellular functions across three major processes: 1) muscle-related (to enrich functions in CMs), 2) extracellular matrix and fibrosis (to enrich functions in Fib, PC, Endo, and vSMCs), and 3) vasculature (to enrich functions in PC, Endo, and vSMCs). For each collection, we defined a collection of search words that were used in MSigDB's search engine with Boolean operators and wildcards. The complete collection of search terms is available in Supplementary File 4. On top of this curation, we included

MSigDB's curated hallmarks for additional gene sets to be enriched in all cell types.

We performed functional enrichment analyses of the MCPs from a cell-type-specific or a multicellular perspective. Briefly, each gene of each cell-type included in the model gets a weight assigned for each MCP, representing its relevance in the definition of the latent variable. For each MCP and cell type, we kept genes with an absolute weight greater than 0 for the rest of the analysis. Given that gene weights are centered at 0, for each MCP and cell type, it is possible to identify two directed gene sets (positive and negative), which represent their levels of expression in non-failing and failing heart tissues, respectively. For each directed gene set, in addition, we identified genes that were relevant in more than a single cell-type and we annotated them as the multicellular component of the program, while the rest of the genes were assigned to their specific cell-type. To test if molecular or cellular functions were overrepresented in each MCP-associated gene set, we performed hypergeometric tests. $P$ value corrections using the BH were performed for each functional gene set collection independently. Overrepresentation of all functional gene set collections was performed for the directed multicellular MCP-associated gene sets. For cell-type-specific gene sets, overrepresentation of functional sets was specific to their expected functions as described before.

Finally, to identify differential multicellular and cell-type-specific functions across MCP1 and MCP2, we estimated the position of functional vectors in the 2D space defined by the MCPs. The functional vector is the result of the addition of enrichment vectors across MCPs and directions (four vectors in total: two MCPs and two directions). To determine the vector of each MCP and direction for each gene set, we multiplied the fraction of genes of the MCP-associated program that overlapped with the functional gene set by the complement of the $p$ value of the hypergeometric test of enrichment. These scores prioritize well-represented and enriched sets in each direction. Addition of the enrichment vectors was done for all functional gene sets with an adjusted $p$ value lower than or equal to 0.1 in at least one MCP-associated gene set. For visualization purposes, representative pathways were manually selected.

## Multicellular coordination networks

MCPs describe global coordination across cell types, however, they do not explicitly describe the degree of dependency of each pair of cells. Thus, we inferred from MCP1 a cell-cell dependency network with directed edges that describe to what extent the molecular profile of one cell-type could predict the profile of each other cell-type. First, we generated cell-type signature matrices per patient sample and study, by enriching the cell-type-specific gene weights of MCP1 into the pseudobulk profiles of their respective cell-type. Enrichment was performed with linear models as implemented in decoupleR v2.6.0[68]. Gene weights with an absolute value lower than 0.1 were excluded from the signature to be enriched. To estimate the weight of the incoming edges of the multicellular coordination networks, we built linear mixed models to predict the signatures of a specific cell type with the signatures of the rest. In these models, the study of origin was treated as a random effect. For a given predicted cell-type, the coefficient estimate of each predictor cell-type multiplied by the overall fit of the model was used as the final edge weight. Edges with a weight <0.2 were filtered out.

## Ligand-receptor analysis

We performed ligand-receptor analysis to aid the interpretation of the MCP1. Thus, as input for this analysis, we used the gene loadings of the MCP1 to extract possible active ligand-receptor pairs, as well as ligand to target gene associations. First, we extracted curated ligand-receptor pairs with the liana v0.1.13 R-package[48]. To calculate ligand–receptor interactions, we extracted the expressed ligand and receptor genes with an absolute gene loading >0.1 in MCP1 for each cell type pair and

summed both loadings to a ligand-receptor interaction score. To connect these ligands with possible downstream targets, we then used the updated Nichenet resource[32]. In detail, for a given pair of cell type A and cell type B, we selected all ligands expressed in cell type A and predicted target genes as defined by the top 10% most extreme gene loadings of the MCP1 from cell type B using the Nichenet algorithm. Background genes were defined as the 30% genes with MCP1 loadings closest to 0. Then, the corrected AUPR was calculated as implemented in nichenetr v.2.0.4. R-package and together with L-R interaction score used for ligand prioritization.

### Experimental corroboration of fibroblast ligands

Neonatal rat ventricular cardiomyocytes (NRVCMs) were isolated from 1- to 2-day-old Sprague-Dawley rat pups using a modified protocol adapted from Miltenyi Biotec's Neonatal Heart Dissociation Kit (mouse/rat, 130-098-373) and Neonatal Cardiomyocyte Isolation Kit (rat, 130-105-420). Briefly, hearts were dissected in ice-cold ADS buffer (1163.6 mM NaCl, 197.2 mM HEPES, 94.2 mM $NaH_2PO_4$, 55.5 mM glucose, 53.6 mM KCl, 8.3 mM $MgSO_4$, pH 7.4). After removal of connective tissue, hearts were minced and enzymatically dissociated in a gentleMACS Octo Dissociator (37 °C, 56 min) using a two-step enzyme mixture (enzyme Mix I contains Enzyme P and Buffer X; enzyme Mix II contains Enzyme A, Enzyme D, and Buffer Y). The cell suspension was filtered through a 70-μm Cellstrainer, centrifuged, and subjected to red blood cell depletion using Anti-Red Blood Cell Microbeads. Cardiomyocytes were purified via magnetic-activated cell sorting (MACS) columns, resuspended in DMEM/F12 medium supplemented with 10% fetal calf serum (FCS) and 1% penicillin-streptomycin-glutamine, and counted using trypan blue exclusion. NRVCMs were serum starved for 48 h before treating them with either Bone morphogenetic protein 4 (BMP4, 10 ng/ml, Thermofisher with Catalog # PHC9531), Neuregulin 1 (NRG1, 10 ng/ml, MedChemExpress with Catalog # HY-P702533), Matrix remodeling associated 5 (MXRA5, 10 ng/ml, MyBioSource with Catalog # MBS1429315), or Phenylephrine (PE, 50 μM, Sigma-Aldrich, Catalog # P6126-10G) in serum-free DMEM for 24 h. Vehicle-treated cells were used as controls.

Total RNA was isolated from NRVCMs using TRIzol reagent (Invitrogen), following the manufacturer's instructions. One microgram of DNA-free RNA was then converted into complementary DNA (cDNA) using the LunaScript First-Strand cDNA Synthesis Kit (New England Biolabs GmbH). Quantitative real-time PCR (qRT-PCR) was conducted using EXPRESS SYBR Green ER reagent (Life Technologies, Inc.) on a LightCycler 480 II system (Roche). The qRT-PCR protocol included an initial denaturation step at 95 °C for 10 min followed by 40 amplification cycles of 15 seconds at 95 °C and 60 seconds at 60 °C. Gene expression levels were normalized to Rpl32, which served as the internal reference gene. Primer sequences are supplied in Supplementary File 9. All NRVCM experiments were performed in six replicates and repeated three times to ensure reproducibility. Statistical testing for differential expression analysis between each treatment condition and controls was performed with linear mixed models, where gene expression was modeled by the fixed effect of the condition (control or treatment) and a random effect of the experiment. $P$ values were adjusted using the BH procedure.

### Study the projection of supporting studies

Supporting single-cell studies were projected into the latent space formed by the MCP latent space estimated from the core studies using the gene weights of the MCPs. To project a new study, we computed from the reference dataset the Moore-Penrose generalized inverse[66] of the concatenated feature weights across cell-type views and multiplied it by the multi-view data of a test patient cohort. The inverse matrix of the feature weights was calculated using the ginv() function from MASS v7.3-57. The multi-view data of the test patient cohort was

centered, where each feature had a mean equal to 0. Associations of clinical covariates or patient conditions with MCP1 and MCP2 were tested in individual and combined models. Pseudobulk expression matrices of projected mouse studies (AngII and TAC) were translated into human gene symbols using biomaRt v2.58.0. We filtered out genes with ambiguous mapping, including duplications. To project HF patient samples of studies without a healthy reference into the core MCP latent space, we first normalized their pseudobulk expression data using an independent healthy heart single-cell human cell atlas[69]. Processing of the healthy single-cell atlas followed identical processing and normalization as the core and supporting studies. The gene expression data of the studies to be projected were then standardized relative to the healthy atlas. For each gene, we subtracted the mean expression of its corresponding gene in the healthy atlas and divided it by the healthy standard deviation. To evaluate that the projected HF samples were located appropriately in the MCP latent space, we created a decision boundary to classify failing from non-failing tissue samples using the information from the core studies. We trained a logistic linear regression to predict HF using the patient values of MCP1 and MCP2 across the four core studies. We then predicted for the projected samples their disease status and calculated the classification error. Association of MCP1 projected scores with available clinical data was done with Pearson correlations and analysis of variance (ANOVA) for continuous and categorical variables, respectively. Multiple testing correction was performed as before with a BH procedure.

### Mapping MCPs to a collection of bulk transcriptomics data

To test if MCPs were traceable in bulk transcriptomics data, we calculated the enrichment of cell-type-specific signatures of each MCP into each study of our bulk transcriptomics data curation. For a given bulk sample within a study, we used decoupleR v2.6.0 to estimate enrichment scores of each cell-type signature using linear models. The gene expression of each study was centered and scaled across samples before the enrichment estimation. AUROCs for each bulk study and cell-type were calculated to test if the estimated cell-type-specific signatures were discriminative of failing and non-failing tissue patient samples. Enriched signatures were sorted by their value, and non-failing samples were used as response variables to calculate AUROCs.

### Annotation of deregulation processes of bulk data

To integrate the observations of gene deregulation in HF from bulk and single-cell transcriptomics data, we explored to what extent a gene reported to be increasing or decreasing its levels of expression in bulk could be explained by changes in tissue composition or by cell-type-specific/multicellular regulatory processes (here referred to in general as molecular processes). We assumed that a change in expression associated with changes in tissue compositions could be traced in genes that have specific expression in cell types that showed a change in composition in HF. Marker genes of one or more cell types could potentially be annotated in this way. In contrast, we assumed that a change in expression associated with a purely molecular process would be reflected by an agreement between the direction of deregulation in bulk data with the overall change across cell types, as reflected by our MCP1. Following these assumptions, we devised two scores from the core single-cell studies that reflect both compositional and molecular processes. The compositional score was calculated for all marker genes of all cell types as a product between the mean log fold change of marker expression across studies (Section 4.3) with the mean t-value across studies of the compositional differential analysis of their specific cell type (Section 4.4). For genes that were markers of more than a single cell type, we summed the scores. The molecular score of all genes modeled with multicellular factor analysis was the mean gene weight across cell types for MCP1, given its proven traceability in bulk. Finally, every gene in the bulk consensus signature was annotated to be potentially deregulated from a compositional,

molecular, compositional and molecular or unknown processes, depending on the agreement between the mean $t$ value of the change in expression between failing and non-failing hearts in bulk transcriptomics datasets and the compositional and molecular scores estimated from the core single-cell studies. If genes had identical signs in the bulk log fold change and the single-cell score, they were annotated with the compositional, molecular, or compositional and molecular label. Genes not profiled with single-cell data or not agreeing with any score were considered unknown.

## Bulk cell type deconvolution

We performed bulk deconvolution to i) test the impact of cell type marker regulation in HF and ii) to infer compositions in bulk RNAseq datasets to identify conserved compositional changes in HF. We used CIBERSORT[70] to estimate cell type compositions in mixed cardiac samples. This method requires a signature matrix that contains marker genes for each cell type. To design this matrix, we selected consensus cell type markers derived from our core HF-studies by selecting markers with an adjusted Fisher $p$ value $< 10e^{-}50$. We then transferred the annotations of bulk deregulation to these markers and identified four subsets of cell type markers: i) unregulated markers, ii) generally deregulated markers, iii) molecularly deregulated markers, and iv) compositionally deregulated markers. We used a healthy cardiac single-nucleus reference data set[1], which was filtered for cells from LV location, and manually aligned to our vocabulary of the seven major cell lineages, then transformed to pseudobulks and finally normalized to transcripts per million (TPM) and kept in linear scale, to build the signature matrix according to the authors' recommendations[70]. We sampled without replacement 30% of all cells to generate the TPM expression values, generating three different signature matrices, which were merged after deconvolution by taking the average predicted compositions. For the benchmarking of the signature matrices, we selected the four core and six supplemental HF sc-studies and summed gene counts for each patient across all cell lineages to get mixtures with known cell type proportions and applied the same normalization procedure as for the signature matrices. The estimated proportions were then evaluated by calculating root-mean-squared-error and pearson's correlation per study. To evaluate the estimated proportions, we performed differential compositional analysis as described in 4.4. We assigned classes of compositional increase, decrease ($t$ test $p$ value $< 0.05$), or no change ($t$ test $p$ value $\geq 0.05$) and calculated F1-scores for each class. For this evaluation of compositional change, we only tested seven of the ten studies, as Amrute2023, Nicin2022, and Liu2022 shared the control samples with the healthy reference.

The HF core bulk studies were then deconvoluted with the compositional signature matrix, and compositional changes were again evaluated as described in 4.4. Cell-type composition matrix from the single-nucleus and bulk cohort was then analyzed together via PC analysis, while each PC was tested for association with HF, technology, or study variable, and explained variance of significantly associated PCs (linear model $p < 0.001$) was summed for the total estimate of associated variance.

## Single-cell integration of fibroblasts

We integrated fibroblast single-cell data from the four core HF studies. To ensure that the single-nucleus data closely resembled the pseudobulk data and the derived MCP, we applied only minimal additional filtering. We retained cells with 200–2500 detected genes and <1% mitochondrial gene content. Genes expressed in at least 10 cells were retained, and counts were log-normalized. Before integration, we removed the top 200 lineage markers from non-fibroblast cell types to reduce background contamination, after which highly variable genes were identified. The top 2000 highly variable genes, common to multiple data sets, were selected for integration. In this feature space, we regressed out total counts per cell, scaled the data, and performed

PCA. This PCA was then adjusted for the sample label by Harmony[60]. In the resulting embedding, we calculated a nearest neighbor graph and performed Leiden clustering to identify cell states. We detected a cluster with low gene counts and expression of cardiomyocyte marker TTN or RYR2, which did not display enrichment of the fibroblast program captured by the MCP1. We removed these cells and repeated the integration steps, resulting in a final integrated fibroblast atlas. We calculated state markers with Wilcoxon tests implemented in SCANPY v1.9.5[71]. We performed differential abundance testing of cell states between conditions as described in 4.4.

## State characterization and division of labor in fibroblasts

We summed counts per patient and per cell state to pseudobulks, which were filtered and normalized with TMM implemented in edgeR[72]. We enriched prior knowledge gene sets from CytoSig[73], PROGENy[74], and MSIG DB[67]. We scaled enrichment scores and selected variable genesets (ANOVA $p$ value $< 0.05$) to apply $t$ tests in one vs. the rest manner to determine significantly upregulated gene sets after BH correction. We selected the top 200 cell state markers based on Wilcoxon $p$ value and enriched them in the gene loading vectors of MCP1 and MCP2. We selected the mean t-statistics of studies calculated from the differential abundance testing of cell states and fit a linear model to characterize the association between state marker enrichment in the MCPs and compositional changes. To characterize the expression pattern of top genes from MCP1 and 2 in different cell states, we enriched the normalized cell state pseudobulks for the top genes from MCP1 and MCP2 (absolute loading >0.1) using the loadings as weights. For this analysis, we multiplied the weights by −1, such that positive values would associate with HF. To characterize the variability of enrichment scores between cell states and patient cohorts, we fit a linear mixed model for each MCP with the formula:

$$ES \sim HF + CS + (1|Study + Patient\_ID)$$

where ES represents the enrichment score, HF represents HF status, and CS represents cell state. Random intercepts were included for both Study and Patient_ID. We then extracted semi-partial $R^2$ values for the fixed effects HF and CS with the r2glmm v0.1.2 R-package[75] to compare the expression pattern across cell states and HF status. To determine an individual gene's expression pattern, the normalized gene expression value was modeled instead of the enrichment score. We assigned division of labor groups based on a combination of semi-partial $R^2$ values: specialist ($R^2$ cell state >0.1 AND $R^2$ HF < 0.1), generalist ($R^2$ cell state <0.1 AND $R^2$ HF > 0.1), and acquired generalist ($R^2$ cell state >0.1 AND $R^2$ HF > 0.1). These gensets were further separated into HF or non-failing associated by using the sign of the gene's loading in MCP1. The MCP gene loadings and the consensus bulk signature were then used as directed rankings and enriched for the division of labor gene sets. All enrichment analyses were performed via *run_ulm()* from the decoupleR v2.7.1 R-package[68].

## Data visualization

All box plots display the minimum, first quartile (Q1), median, third quartile (Q3), and maximum, with outliers plotted separately as points beyond 1.5 times the interquartile range (IQR) from Q1 or Q3.

## Use of large language models

Publicly available large language models (Github Copilot, ChatGPT) were used to generate code and review the manuscript for grammar and spelling, every output was manually reviewed and manually edited if necessary.

## Reporting summary

Further information on research design is available in the Nature Portfolio Reporting Summary linked to this article.

## Data availability

The single-nucleus RNAseq and bulk RNAseq used in this study are available in public databases under different accession codes, summarized in Supplementary File 1. The processed transcriptomic data are available on Zenodo[76] (https://doi.org/10.5281/zenodo.15261569). Source data are provided with this paper. The experimental data generated in this study are provided in the Source Data file. Source data are provided with this paper.

## Code availability

All code associated with this publication is available at github.com/saezlab/reheat2_pub[77] and at github.com/saezlab/reheat2_shiny.

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

## Acknowledgements
R.O.R.F. acknowledges the support of the German Science Foundation (DFG) through the CRC1550 Molecular Circuits of Heart Disease. JDL acknowledges the support of the German Federal Ministry of Education and Research (Bundesministerium für Bildung und Forschung) through CureFib. Supported by the DZHK (German Center for Cardiovascular Research), funding code: 81 × 4500130. J.L.B. is supported by the Galician Government through the fellowship ED481B_072. This work is partly supported by CRC1550 (DFG, #464424253) to J.S.R. and N.F. A.Y.R. and N.F. are funded by the German Center for Cardiovascular Research (DZHK partner site project). We thank Leonie Küchenhoff and Ines Rivero for critical feedback on the manuscript. Importantly, we acknowledge all data authors that are cited in this study. The authors gratefully acknowledge the data storage service SDS@hd supported by the Ministry of Science, Research, and the Arts Baden-Württemberg (MWK) and the German Research Foundation (DFG) through grant INST 35/1503-1 FUGG.

## Author contributions
J.D.L. and R.O.R.F. data curation, formal analysis, investigation, methodology, resources, software, writing—original draft preparation. J.L.B.: formal analysis, validation, writing—revision. M.S.: investigation. A.Y.R.: supervision, resources. N.F.: supervision, funding acquisition. J.S.R.: supervision, project administration, funding acquisition, writing—revision.

## Funding

## Competing interests
J.S.R. reports funding from GSK, Pfizer, AstraZeneca, and Sanofi, and fees/honoraria from Travere Therapeutics, Stadapharm, Astex, Pfizer, Grunenthal, and Owkin. The remaining authors declare no competing interests.
