## [Transparent Peer Review file · Nature Communications]

A cross-study transcriptional patient map of heart failure defines conserved multicellular coordination in cardiac remodeling

Corresponding Author: Professor Julio Saez Rodriguez

Version 0:

Reviewer comments:

Reviewer #1

(Remarks to the Author)

In this report by Lanzer et al, a cross-study analysis of cardiac transcriptomic reprogramming in failing hearts was conducted by integrating 25 published studies with bulk-RNA-seq and Sn/Sc RNA seq datasets from human heart failure and non-heart failure cohorts. These datasets were evaluated to determine molecular conservation, cellular composition (among 7 cell types) and multicellular molecular signatures across different studies. Following the initial evaluation, authors focused on single-nucleus studies and demonstrated cellular composition changes and cell-type specific molecular reprogramming in heart failure across multiple studies. From these analyses, a joint multicellular patient map was constructed based on multicellular factor analysis and 10 multicellular programs were derived, including two major ones that differentiate heart failure from the non-failing hearts. Cell-cell interactions were further elucidated based on ligand-receptor pairing. Fibroblasts were selected for further analysis due to its major contributions to the cellular composition change and cell-myocyte interactions. Subclusters of fibroblasts were identified to maintain specific cellular functions during reprogramming in heart failure. Finally, the multi-cellular program identified from sn-RNA datasets was further evaluated in bulk RNA-seq transcriptome datasets. While the molecular changes identified from multicellular program were found to be conserved in the bulk transcriptome data, cellular composition changes were estimated from bulk transcriptome profiles. Combined all these results, authors tested the robustness of this multicellular program model in independent datasets, and identified specific signature in cardiomyocytes and fibroblasts associated with heart failure as well as post-LVAD recovery.

Different transcriptome analyses tools have been developed and the current study offered a model from a very large cohort of snRNA and bulk RNA seq samples. While it is perplexing that model failed to detect significant molecular signature associated with age, sex, and other well established comorbidities, the findings from this report, largely based on human heart datasets, particularly on the cell type specific changes, are interesting and potentially useful for clinical stratifications, diagnosis and treatment. The overall informatic pipeline was well articulated and logically assembled. While individual approaches may not be that novel, the study has significant merits in systematic integration and rigorous testing of the model of multicellular program (MCP).

1. It is perplexing that MCP developed in this study could not identify contributions from specific comorbidities, including sex, age, metabolic status and hypertension, as well as clinically well-defined etiologies (such as HCM vs. DCM, vs. ischemic, HFrEF vs. HFpEF). Is this a limitation of the model?
2. The identified cell-cell communication between cardiomyocytes and fibroblasts as dominant interactions was interesting and also a source of concern. It is not clear if authors can re-analyze cell-cell communications based on directionality of such interactions. From the current analysis, it appears that fibroblasts have the most significant impact on cardiomyocytes. It would be interesting to find out which cell types have the most significant impact on fibroblasts?
3. Among the genes identified to be associated with heart failure and post-LAVD recovery, it is not clear whether other clinic features (such as severity of heart failure, remodeling, injury sizes and so on) are also considered and analyzed. At this stage, it is not feasible to establish any causal relationship to the clinic outcome.
4. Authors claimed the new multicellular program can provide new mechanistic insights to the pathogenesis of heart failure. It is not clear that other than established paradigm in ECM remodeling, inflammation, metabolic reprogramming, and several pathology associated signaling pathways, what are the NEW mechanism being revealed from this study. Please elaborate more clearly.
5. To provide better access and utility of this toolset for researchers and clinicians, authors should provide better description to their developed platform and its applications, perhaps as an additional supplemental information.

(Remarks on code availability)

I can't comment on the reproducibility of the results.

Reviewer #2

(Remarks to the Author)

The manuscript presents a meta-analysis on the transcriptomic patterns of Heart failure (HF) and non-failing (NF) patients. By integrating multiple studies from both sn- and bulk- RNA sequencing profiles, this study depicts a HF patient map by analyzing the cell-type composition, cell-cell coordination & communication, multicellular programming, and labor division within fibroblasts.

In all, I find the topic of this study novel and meaningful. The study employs various computational techniques; however, the authors do not provide sufficient methodological details or adequately interpret the output results, making the manuscript difficult to follow. The manuscript would benefit from additional clarification and greater rigor. Specific comments are as follows:

Major comments:

1. The multicellular program (MCP) is one of the core concepts throughout this study. Although it has been briefly introduced with proper citation, I find it necessary to clearly describe this concept using both intuitive illustration and rigorous definitions. Otherwise, readers may not understand what this means and how it is obtained without prior knowledge on the author's previous work MOFA.

2. In this study, cell annotation for single-cell datasets is performed individually for each dataset. However, this approach may make it more challenging to compare cell types across studies. It is not clear how the cell types are aligned across different studies, as they might use different standards of cell type annotation and different sets of cell type terminology.

Another alternative approach is to integrate multiple datasets, identify unified clusters and annotate these clusters. The authors need to justify the choice of their annotation strategy.

3. The manuscript uses technical jargon that may limit accessibility to a broader audience. For example, "However, transcriptional directionality was conserved, as shown by cross-study enrichment analysis (upregulated mean enrichment score 0.58, downregulated mean enrichment score -0.54, Supplementary Figure 2B)," what does the upregulated/downregulated mean enrichment score mean? How strong are these scores? What is the statistical significance (e.g., p-value, FDR) of these enrichment scores?

Also, how is the co-expression network defined in Section 2.4? It is confusing that the authors claimed "a consistency between the number of ligand-receptor interaction and co-expression network" but "a weak correlation between ligand-receptor coexpression and gene expression coordination".

4. The authors need to provide rigorous description of how the projection of new dataset into the patient map. For example, what's the formula to calculate the MCP coordinates for a new dataset, given the current patient map?

5. How to explain the low overlap of the differentially expressed genes among the bulk studies?

6. In detail, how was NicheNet performed when the input contained multiple snRNA-seq datasets from multiple samples? How does the cell-cell communication pattern vary across different studies within the HF group?

7. What are the mostly altered ligand-receptor pairs and ligand-target pairs between HF and NF samples? The visualization of the cell-cell communications is not intuitive enough (especially the ligand-receptor pairs). I suggest using chord or circled diagrams connecting ligands and receptors to further visualize the dominant L-R pairs given the sender and receiver cell clusters.

8. The multicellular information network of NF hearts (like Figure 3E) is needed for a clear comparison.

Minor comments:

1. Figure 2D is not correctly cited in the main text.

2. How is the term "comparable" defined in Section 2.2.2?

3. "Each data set was processed individually by applying common thresholds, filtering genes with at least 10 cells and cells with at least 200 genes and counts were log-normalized." It's not clear what "common thresholds" mean here.

4. "We hypothesized that if two independent studies using the same technology captured similar molecular processes in HF, a disease classifier trained on one study's data could predict the disease status of heart tissue samples in the other (Figure 2A). This approach allowed us to assess how well independent studies with different technical and clinical characteristics

aligned at various scales.” It’s not clear what the different characteristics are for independent studies using the same technology.

(Remarks on code availability)

Version 1:

Reviewer comments:

Reviewer #1

(Remarks to the Author)

The revision has adequately addressed the concerns raised by this reviewer.

(Remarks on code availability)

I don't have adequate expertise to evaluate the code function.

Reviewer #2

(Remarks to the Author)

No more comments.

(Remarks on code availability)

The images or other third party material in this Peer Review File are included in the article’s Creative Commons license, unless indicated otherwise in a credit line to the material. If material is not included in the article’s Creative Commons license and your intended use is not permitted by statutory regulation or exceeds the permitted use, you will need to obtain permission directly from the copyright holder.

Reviewer #1 (Remarks to the Author):

In this report by Lanzer et al, a cross-study analysis of cardiac transcriptomic reprogramming in failing hearts was conducted by integrating 25 published studies with bulk-RNA-seq and Sn/Sc RNA seq datasets from human heart failure and non-heart failure cohorts. These datasets were evaluated to determine molecular conservation, cellular composition (among 7 cell types) and multicellular molecular signatures across different studies. Following the initial evaluation, authors focused on single-nucleus studies and demonstrated cellular composition changes and cell-type specific molecular reprogramming in heart failure across multiple studies. From these analyses, a joint multicellular patient map was constructed based on multicellular factor analysis and 10 multicellular programs were derived, including two major ones that differentiate heart failure from the non-failing hearts. Cell-cell interactions were further elucidated based on ligand-receptor pairing. Fibroblasts were selected for further analysis due to its major contributions to the cellular composition change and cell-myocyte interactions. Subclusters of fibroblasts were identified to maintain specific cellular functions during reprogramming in heart failure. Finally, the multi-cellular program identified from sn-RNA datasets was further evaluated in bulk RNA-seq transcriptome datasets. While the molecular changes identified from multicellular program were found to be conserved in the bulk transcriptome data, cellular composition changes were estimated from bulk transcriptome profiles. Combined all these results, authors tested the robustness of this multicellular program model in independent datasets, and identified specific signature in cardiomyocytes and fibroblasts associated with heart failure as well as post-LVAD recovery.

Different transcriptome analyses tools have been developed and the current study offered a model from a very large cohort of snRNA and bulk RNA seq samples. While it is perplexing that model failed to detect significant molecular signature associated with age, sex, and other well established comorbidities, the findings from this report, largely based on human heart datasets, particularly on the cell type specific changes, are interesting and potentially useful for clinical stratifications, diagnosis and treatment. The overall informatic pipeline was well articulated and logically assembled. While individual approaches may not be that novel, the study has significant merits in systematic integration and rigorous testing of the model of multicellular program (MCP).

We thank the reviewer for the positive comments and for highlighting the benefit of systematic integration. We share the surprise regarding the weak association of other clinical covariates in the heart failure and recovery transcriptional responses. We added a new supervised modelling approach to study these interactions and confirmed that the lack of interaction between the heart failure signals and the other clinical covariates is a property of the data and, possibly, the disease or disease status we studied.

1.1. It is perplexing that MCP developed in this study could not identify contributions from specific comorbidities, including sex, age, metabolic status and hypertension, as well as clinically well-defined etiologies (such as HCM vs. DCM, vs. ischemic, HFrEF vs. HFpEF). Is this a limitation of the model?

Our results could be both a limitation of the model and of the available data.

Group factor analysis, which we use to construct our multicellular patient map, captures variability across cell types simultaneously. However, the latent variables that define the multicellular programs in our model are constrained by the heterogeneity in gene expression captured within individual studies,

which is often influenced by cohort size. Given the unsupervised nature of our approach, variables with little effects in gene expression variability (e.g. sex or age) would have less likelihood to be associated with any factor, particularly in the meta-analysis setting where we constrain the model for shared variance across datasets.

However, another possibility is that the absence of association between the MCP1 and MCP2 and other clinical covariates reflects lack of variability in gene expression associated with these same covariates. It is important to remark that the patients from the distinct studies suffered from the chronic end-stage of heart failure with potentially very limited variability since they all required transplantation.

To evaluate the limits of our multicellular model and to quantify the expected variability in gene expression associated with sex, age, and etiologies, we complemented our unsupervised analysis by explicitly modeling the variability of gene expression for all genes across cell-types included in our factor analysis. First, we modeled gene expression in terms of an interaction between heart failure and sex or age. Then, we tested for differential expression across etiologies using only heart failure samples.

For sex and age, we observed almost no genes showing interaction with heart failure across all cell-types (mean number of genes with significant interaction (adj. p-value < 0.05) coefficient with age and sex equal to 1.29 and 0.1, respectively). These results suggest that the analysed heart failure data does not capture variability associated with age and sex, and that the lack of association between the activity of multicellular programs with these covariates does not reflect a limitation of our model. Regarding etiologies, we observed only a mean of 8.5 and 37.1 differentially expressed genes across cell-types for HCM and ICM, respectively (adj. p-value < 0.05). Thus, it is likely that this minimal difference between etiologies does not represent a major axis of variation and thus, was not captured by our model. Overall, these results reinforce the idea of convergent molecular responses in end-stage heart failure and extend this observation across different age groups and sexes. Nonetheless, we acknowledge the limitations of the data, including the absence of additional clinical annotations and the limited diversity of patients (e.g., the exclusion of HFpEF cases), which constrain a more comprehensive exploration of the multicellular patient map in relation to other variables of interest.

To highlight these observations we have included a new panel in Supplementary Figure 6 and modified the results and discussion sections. In addition, we provide a supplementary table with the genes related to etiology differences (Supplementary table 3).

Supplementary

Figure

6.

F. Number of genes whose variability in expression can be associated significantly with an interaction between HF and age or sex. Linear mixed models, adjusted p-value < 0.05.

G. Adjusted p-values of associations, using linear mixed models, between MCP1 and MCP2 scores in HF samples with reported clinical variables.

H. Number of genes whose variability in expression in HF patients can be associated significantly with a difference in etiologies. Linear mixed models, adjusted p-value < 0.05.

Result section 2.3:

[...]The multicellular space showed little to no variability associated with body mass index, age, or sex (mean R^2 of 0.3%, 3%, and 0%, respectively). Differential expression analysis testing for interactions between heart failure and sex or age confirmed no significant effects, with a mean of 1.29 and 0.1 genes, respectively, showing a significant interaction coefficient across cell-types (Supplementary Figure 6F).

[...]By fitting linear mixed models to both MCPs using patient information as predictors and accounting for studies as random effects, we found no association between the distribution of HF patients across MCP1 and MCP2 and their organ sampling sites, acquisition mode, primary or secondary causes of HF, or etiology. Differential expression testing across etiologies confirmed the low signal, with a mean of 8.5 and 37.1 differentially expressed genes for HCM and ICM, respectively, across cell types (Supplementary Figure 6H, Supplementary table 3).

Discussion:

The integration of transcriptomics data across scales enabled us to build a patient map that captures heart tissue variability across cell types, based on the activation of two distinct multicellular programs (MCP1 and MCP2). Interpreting this variability with clinical covariates could help identify molecular markers of heart disease. We showed that MCP1 and MCP2 successfully distinguish failing from non-failing hearts across core studies. Although the patient map did not show strong associations with expected clinical covariates such as age, sex, or etiology, our analyses suggest that this reflects limited variability in these covariates within the collected patient cohorts rather than a limitation of the modeling strategy. Despite this, we propose that the patient map has translational potential as a reference for analyzing single-cell data from related HF contexts. By projecting independent datasets onto MCP1 and MCP2, we facilitated the reinterpretation of studies on fetal samples, myocardial infarction, animal models, and LVAD-treated patients.

Details of the modeling strategies can be found in a new Methods section 4.10

4.10 Supervised analysis of gene expression of heart failure patients

To complement our unsupervised multicellular factor analysis, we examined associations between gene expression variability in heart failure patients and sex, age, and etiology. For sex and age, we fitted linear mixed models to all samples, incorporating an interaction term between heart failure and either sex or age, with studies treated as random effects. To assess differences across etiologies, we applied a linear mixed model to heart failure samples, using etiology as a fixed effect and study as a random effect. Due to sample size constraints, only DCM, HCM, and ICM samples were included. DCM samples were used as the reference intersect in the model. Additionally, gene expression in heart failure samples was centered and standardized relative to non-failing samples ($(\text{expression} - \text{mean}^{\text{NF}})/\text{SD}^{\text{NF}}$) with centering performed independently within each study. P-values were adjusted for multiple testing using the BH procedure.

2. The identified cell-cell communication between cardiomyocytes and fibroblasts as dominant interactions was interesting and also a source of concern. It is not clear if authors can re-analyze cell-cell communications based on directionality of such interactions. From the current analysis, it appears that fibroblasts have the most significant impact on cardiomyocytes. It would be interesting to find out which cell types have the most significant impact on fibroblasts?

We apologize for the confusion related to our communication analysis. In response to the Reviewer 2's comment 2.3, we have clarified the communication analysis by including a new schematic (new Supplementary Figure 7) and an extended description of the methodology in our methods section (see answer 2.3).

Briefly, first, we calculated expression scores of the MCP1 per cell type and patient. Then, to build the network of cell-type coordination, we used linear mixed models to assess how well the MCP1 gene expression profile of a cell type explains the gene expression of other cell types. The directionality of this network reflects the importance of a cell to predict (outgoing) or be predicted by (incoming) another cell-type via the linear regression model. The reconstructed co-expression network of MCP1 is then used to constrain the inference of cell-cell communication via ligand-receptor interactions as a means to potentially reduce false positives, which is a major limitation of these approaches as our previous benchmarks suggest (Dimitrov *et al*, 2024, 2022). These representations of multicellular coordination can be exploited to nominate potential regulators of the gene expression of distinct cell-types during cardiac remodeling. And, as suggested by the reviewer, we have added supplementary analyses studying the regulators of fibroblasts.

First, we compared cell types for their predictive importance of fibroblasts, which are represented by the incoming edges to the fibroblast node in the network in main figure 3E. For convenience, we plotted here the predictive importances for fibroblasts separately for HF and NF networks.

Revision figure. Comparing the importances of different cell types to predict the fibroblast specific program within the multicellular program 1 (MCP1).

In the non-failing heart, the fibroblast program was predicted by endothelial and myeloid cells, and pericytes. In contrast, in the failing heart in addition to myeloid cells, we observed an increased predictive importance of cardiomyocytes. While the crosstalk between fibroblasts and myeloid cells in heart failure is known to drive fibrotic tissue remodeling based on previous studies (Amrute *et al*, 2024; Hoeft *et al*, 2023) the regulatory interaction between cardiomyocytes and fibroblasts appears to be an

emerging cross-talk between both cell types, based on the difference in the predictor importance of cardiomyocytes in the non-failing and failing heart coordination network. Thus, we extracted possible ligands for both CM and Myeloids targeting Fibroblasts and highlighted those in the manuscript.

We report these findings now in the results section 2.4:

[...]These ligands could be key regulators of coordinated multicellular fibrotic processes in HF and hence potential modulators of myocardial remodeling. In HF, fibroblasts were best predicted by CMs and Myeloids. The inference of potential ligands from these cell types regulating fibroblast gene expression (Supplemental Figure 8E,F) identified a set of ligands previously linked to have pro-fibrotic effects, including LGALS9 and PDCD1LG2 from Myeloids and CALR and SEMA3E from CMs.

Updated Supplemental Figure 7.

E-F) NicheNet results for the top ligand-receptor pairs between Myeloid and Fibroblasts (E) and CM and fibroblasts (F). NicheNet derived a regulatory potential (corrected AUPR, x-axis) of a given ligand to deregulate a gene signature which here represents extreme gene loadings of the target cell type taken from the MCP1. Ligand-receptor (L-R) score (y-axis) represents the mean gene loadings of the ligand and the receptor of a given cell type pair. If a ligand connected with multiple receptors, the median L-R score was calculated. Color represents the number of cell types that express this ligand in HF.

We propose that our generated resource can be leveraged to study and query the different cell type interactions, as requested by the reviewer. Motivated by this comment, we added a new interactive module to our deployed web platform (<https://www.saezlab.shinyapps.io/reheat2/>). The "Multicellular Network" module now allows users to explore the failing and non-failing networks interactively. For any selected cell-type pair, users can examine predicted ligand-receptor interactions as well as ligand-

target gene associations. Additionally, in response to Comment 5, we have included a supplementary note providing a detailed description of this new functionality.

3. Among the genes identified to be associated with heart failure and post-LAVD recovery, it is not clear whether other clinic features (such as severity of heart failure, remodeling, injury sizes and so on) are also considered and analyzed. At this stage, it is not feasible to establish any causal relationship to the clinic outcome.

We did not use any additional clinical data during the associations of MCP1 with post-LVAD recovery, and we agree that our association analyses can not lead to a causal relationship to the clinic outcome. To evaluate if and which clinic features of recovered and not recovered patients could be associated with post-LVAD recovery, we performed statistical tests at the multicellular program and gene level using the extensive clinical data provided in the Amrute *et al.* publication (Amrute *et al.*, 2023).

First, we assessed the relationship between MCP1 scores and clinical variables in DCM patients prior to LVAD implantation using analysis of variance (ANOVA) for categorical variables and Pearson correlation for continuous variables, finding no significant associations (adjusted p-value < 0.05, Review Figure 2). These results indicated that the MCP1 scores are not driven by a recorded clinical covariate that could act as a confounder of the association with the recovery post LVAD.

Review Figure 2. Adjusted p-values of association of MCP1 and clinical variables for all tissue samples of DCM patients before the implantation of LVAD. Clinical covariates separated by the type of test fitted: Pearson correlation for continuous variables and ANOVAs for categorical variables.

In addition, we tested for association between clinical covariates and cell-type specific gene expression. For this we used the results of differential expression analysis between recovered and non-recovered patients shown in Supplementary Figure 13. We collected all genes with differential expression that agreed with the direction of their gene weight in MCP1, and performed association tests with clinical data with Pearson correlations and ANOVAs as described before. Similarly, we found no significant associations (adjusted p-value < 0.05) between gene expression and clinical data. These results suggest that MCP1 captures treatment response-related information that is independent of the measured clinical variables. If the fibrotic and low inflammatory tissue state of the patients (based on their location in

MCP1) is related to the cause of treatment effectiveness is something that we agree can not be established with our data. However, we believe that this result highlights the importance of our reference to build mechanistic hypotheses of cardiac remodeling that could impact treatment.

We have modified the manuscript to highlight these results and moderate our claims between the association of MCP1 activation and recovery.

Results section **2.8 Reinterpreting independent data with the multicellular program of heart failure**

[...] Heart tissue samples from recovered and non-recovered dilated cardiomyopathy (DCM) patients, before and after LVAD implantation, also aligned with MCP1 activation (Figure 6E). Among recovered patients, we observed a significant change in MCP1 activation after LVAD implantation, where post-implantation samples resembled those from non-failing hearts (paired t-test, adj. p-value = 0.003). Pre-implantation samples also showed differences in MCP1 activation between recovered and non-recovered patients (t-test, adj. p-value = 0.012), indicating that this multicellular program reflects disease severity and can offer insights into recovery potential. Comparison of gene expression between recovered and non-recovered patients before implantation revealed alignment with MCP1 primarily in endothelial cells, fibroblasts, and myeloid cells. In all three cell types, we observed coordinated downregulation of inflammatory responses, including cytokines (IL17a, IL2, IL15, TNF α) and growth factors (EGF) (Figure 6F). These results suggest that lower inflammation levels may be linked to improved recovery in HF patients post-LVAD. We evaluated whether clinical features were associated with MCP1 activation levels and the expression of related genes between recovered and non-recovered patients. No significant associations were found, suggesting that MCP1 captures treatment response-related information that is independent of the measured clinical variables.

We have expanded the methods section (4.14) to explain these analyses.

[...] Association of MCP1 projected scores with available clinical data was done with Pearson correlations and analysis of variance (ANOVA) for continuous and categorical variables, respectively. Multiple testing correction was performed as before with a BH procedure.

4. Authors claimed the new multicellular program can provide new mechanistic insights to the pathogenesis of heart failure. It is not clear that other than established paradigm in ECM remodeling, inflammation, metabolic reprogramming, and several pathology associated signaling pathways, what are the NEW mechanism being revealed from this study. Please elaborate more clearly.

We thank the reviewer for this comment, as indeed it was not clearly articulated which new insights our approach can provide in the previous version of the manuscript. While our study does not aim to uncover entirely novel mechanistic paradigms, we take a tissue-level approach to infer multicellular programs that enabled us to reconstruct previously undescribed models of cellular coordination, offering a novel systems-level view of heart failure.

We believe that this systems-level view of heart failure serves as a valuable platform to guide mechanistic investigations. To illustrate this, we incorporated new experimental validation. Specifically, we investigated three ligands (BMP4, MXRA5, and NRG1) that were predicted to mediate the emerging fibroblast-to-cardiomyocyte communication in HF. Using neonatal rat ventricular cardiomyocytes (NRVCMs), we assessed their effects on key markers of stress, hypertrophy, and

fibrosis. BMP4 and NRG1 increased expression of *Nppa*, *Nppb*, *Col1a1*, and *Tgfb1*, while MXRA5 induced *Nppa* and *Tgfb1*, and repressed *Nppb*.

We further applied NicheNet to identify putative downstream targets and assessed their expression changes upon ligand treatment. This enabled us to establish novel ligand - target relationships between fibroblasts and cardiomyocytes. These experiments demonstrate how our resource can inform hypothesis generation and experimental prioritization. The analytical workflow used here is accessible via our updated public web application (see comment 1.5) and can be extended to any interaction in the predicted coordination network.

Thus, besides confirming known pathogenic pathways, our study provides a refined map of their orchestration in the failing heart. We believe this systems-level integration represents a useful framework for future mechanistic explorations, as illustrated in the example above. We have added these new data to the manuscript (new Supplementary Note 4, new Supplementary Figure 9) and revised the manuscript accordingly.

From Results section 2.4

Additionally, ligands like BMP4 and NRG1 were identified as potential regulators of the CM stress response within MCP1 (e.g., ANKRD1, CCND1) (Figure 3G). BMP4 is known to play a role in HF, particularly in cardiomyocyte transdifferentiation³⁹, while NRG1 has been implicated in cardiomyocyte division and migration⁴⁰. We corroborated experimentally in cell cultures of neonatal rat ventricular cardiomyocytes (NRVCM) the gene regulatory effects of BMP4, NRG1 and MXRA5 on CM target genes like canonical hypertrophy (NPPA, NPPB) and fibrosis (TGFB1, COL1A1) marker (Supplementary Figure 9A, B), as well as on predicted target genes within MCP1 (e.g. ANKRD1, CCND1, IFNAR2, NAV2, MCL1, SACS, SOCS2) (Supplementary Figure 9C, Supplementary Note 4). Thus, these ligands could be key regulators of coordinated multicellular fibrotic processes in HF and hence potential modulators of myocardial remodeling.

[...] In summary, we reconstructed a multicellular blueprint of cell-cell interactions in HF, revealing the coordinated processes captured by MCP1 and their potential ligand-receptor mechanisms (Supplementary Table 5). Our analysis highlights the central coordinating role of fibroblasts in HF, particularly the increased communication between fibroblasts and cardiomyocytes, supported by predictive models of ligand-receptor interactions and cellular coordination, as well as by experimental validation of selected ligands in vitro.

In a new supplementary note and supplementary figure, we describe the results in greater detail:

Supplementary Note 4: Experimental investigation of fibroblast derived ligands in cardiomyocytes.
*To validate our computational predictions of the gene regulatory effects of fibroblast's ligands in CMs during HF, we selected three ligands expressed by Fibs (BMP4, MXRA5, and NRG1) and investigated their pro-hypertrophic effects in cell cultures of neonatal rat ventricular cardiomyocytes and the changes in gene expression they generated (Methods). BMP4 and NRG1 treatment upregulated the cardiomyocyte stress response markers *Nppa* and *Nppb* (Supplementary Figure 9A-B adj. p-value of t-test from linear mixed model < 0.05), supporting their involvement in hypertrophic signaling as predicted by our cell-cell communication analysis using gene expression data of MCP1. In contrast, MXRA5 upregulated *Nppa* but downregulated *Nppb* (adj. p-value of t-test from linear mixed model < 0.05), indicating a distinct regulatory mechanism than the one inferred in silico (Supplementary Figure*

9A-B). *NRG1* did not affect fibrotic markers *Col1a1* or *Tgfb1*, whereas *BMP4* and *MXRA5* selectively upregulated *Tgfb1* without altering *Col1a1* (adj. p-value of t-test from linear mixed model < 0.05), indicating potential pro-fibrotic roles (Supplementary Figure 9A-B). As expected, phenylephrine (PE), used as a positive control, robustly increased *Nppa*, *Nppb*, *Col1a1*, and *Tgfb1* expression, validating the experimental system (Supplementary Figure 9A–B, adj. p-value of t-test from linear mixed model < 0.05).

To further dissect ligand-specific signaling, we assessed the expression of predicted NicheNet targets within MCP1 following ligand treatment. For this analysis we selected genes with a MCP1 gene loading < -0.2 and a target weight >0 assigned by NicheNet, resulting in a list of 19, 22 and 16 target genes for *BMP4*, *MXRA5*, and *NRG1*, respectively. From these lists we selected ten, nine and nine genes for the respective ligands, based on a literature research to prioritize candidates with a possible functional relevance for hypertrophy. qPCR analysis revealed that *BMP4*, *MXRA5*, and *NRG1* induced the expression of 6 out of 10, 2 out of 9, and 3 out of 9 of their respective selected predicted target genes, respectively (Supplementary Figure 9C, adj. p-value of t-test from linear mixed model < 0.05). From the 11 upregulated genes by *BMP4*, *MXRA5*, and *NRG1*, PE did not induce seven genes despite strongly inducing classical hypertrophy and fibrosis markers (Supplementary Figure 9C), suggesting that the identified ligand-target gene associations are often ligand specific. Among these specific interactions are the upregulation of *NAV2* and *SACS* by *BMP4*, with unknown function in cardiac fibrosis or hypertrophy, but could be functionally involved in cytoskeletal coherence (*NAV2* (Schmidt et al, 2009)) and mitochondrial function (*SACS*). *NRG1*-induced *IFNAR2* and *LYR* could be linked to interferon response (Ninh et al, 2024; Tran et al, 2024) and mitochondrial health (Chen et al, 2016) in cardiomyocytes, respectively, while *MXRA5* might have protective effects on cell survival (*MCL1* (Widden & Placzek, 2021)) and anti-inflammatory effects (*SOCS2* (Esper et al, 2012; Yuan et al, 2024)). Together, these findings corroborate many ligand-target predictions and reveal canonical and non canonical ligand targets involved in cardiomyocyte hypertrophy and fibrosis. These results support the relevance of the selected ligands and their target genes in the multicellular program of HF and underscore the need for mechanistic studies in *in vitro* and *in vivo* models to elucidate their roles in cardiac pathology.

Supplementary Figure 9. Experimental Validation of fibroblast ligands inducing stress and hypertrophy in cardiomyocytes

- A. Neonatal rat ventricular cardiomyocytes (NRVCMs) were treated for 24 h with Bone Morphogenetic Protein 4 (BMP4), Matrix Remodeling Associated 5 (MXRA5), or Neuregulin 1 (NRG1), and analyzed their effect on gene expression by quantitative real-time PCR (qPCR) for markers of hypertrophy (Nppa, Nppb) and fibrosis (Col1a1, Tgfb1). Phenylephrine (PE) treatment was used as a positive control. No treatment was used as a negative control (Ctrl).
- B. Differential expression statistical testing between each NRVCM treated condition and control experiments. Markers of hypertrophy and fibrosis are highlighted
- C. Differential expression statistical testing between each NRVCM treated condition and control experiments. Downstream signaling target genes as predicted by NicheNet are presented.

In all panels, data represent 18 measurements performed in three independent experiments, each with six technical replicates. In B and C, t-values extracted from the t-test of the coefficient of the fixed effect term (condition) from a linear mixed model of each target gene expression, with a random effect of experiments. Stars denote adjusted p-values (BH procedure) below or equal to 0.05. Grey tiles denote not performed tests.

We added also a Methods section 4.14 and supplementary table 9 to describe the experiments in more detail:

4.14 Experimental corroboration of fibroblast ligands

Neonatal rat ventricular cardiomyocytes isolation, culture and ligand treatment

Neonatal rat ventricular cardiomyocytes (NRVCMs) were isolated from 1- to 2-day-old Sprague-Dawley rat pups using a modified protocol adapted from Miltenyi Biotec’s Neonatal Heart Dissociation Kit (mouse/rat, 130-098-373) and Neonatal Cardiomyocyte Isolation Kit (rat, 130-105-420). Briefly, hearts were dissected in ice-cold ADS buffer (1163.6 mM NaCl, 197.2 mM HEPES, 94.2 mM NaH₂PO₄, 55.5 mM glucose, 53.6 mM KCl, 8.3 mM MgSO₄, pH 7.4). After removal of connective tissue, hearts were minced and enzymatically dissociated in a gentleMACS Octo Dissociator (37°C, 56 min) using a two-step enzyme mixture (enzyme Mix I contain Enzyme P and Buffer X; enzyme Mix II contain Enzyme

A, Enzyme D, and Buffer Y). The cell suspension was filtered through a 70- μ m Cellstrainer, centrifuged, and subjected to red blood cell depletion using Anti-Red Blood Cell Microbeads. Cardiomyocytes were purified via magnetic-activated cell sorting (MACS) columns, resuspended in DMEM/F12 medium supplemented with 10% fetal calf serum (FCS) and 1% penicillin-streptomycin-glutamine, and counted using trypan blue exclusion. NRVCMs were serum starved for 48 h before treating them with either Bone morphogenetic protein 4 (BMP4, 10 ng/ml, Thermofisher with Catalog # PHC9531), Neuregulin 1 (NRG1, 10 ng/ml, MedChemExpress with Catalog # HY-P702533), Matrix remodelling associated 5 (MXRA5, 10 ng/ml, MyBioSource with Catalog # MBS1429315), or Phenylephrine (PE, 50 μ M, Sigma-Aldrich, Catalog # P6126-10G) in serum-free DMEM for 24 h. Vehicle treated cells were used as controls.

RNA isolation and qRT-PCR

Total RNA was isolated from NRVCMs using TRIzol reagent (Invitrogen), following the manufacturer's instructions. One microgram of DNA-free RNA was then converted into complementary DNA (cDNA) using the LunaScript First-Strand cDNA Synthesis Kit (New England Biolabs GmbH). Quantitative real-time PCR (qRT-PCR) was conducted using EXPRESS SYBR Green ER reagent (Life Technologies, Inc.) on a LightCycler 480 II system (Roche). The qRT-PCR protocol included an initial denaturation step at 95°C for 10 minutes followed by 40 amplification cycles of 15 seconds at 95°C and 60 seconds at 60°C. Gene expression levels were normalized to *Rpl32*, which served as the internal reference gene. Primer sequences are supplied in Supplementary table 9. All NRVCM experiments were performed in six replicates and repeated three times to ensure reproducibility. Statistical testing for differential expression analysis between each treatment condition and controls was performed with linear mixed models, where gene expression was modeled by the fixed effect of the condition (control or treatment) and a random effect of the experiment. P-values were adjusted using the BH procedure.

5. To provide better access and utility of this toolset for researchers and clinicians, authors should provide better description to their developed platform and its applications, perhaps as an additional supplemental information.

We agree with the reviewer and added a new Supplementary Note 8 and Supplementary Figure 14 to the manuscript, explaining the functionality of the ReHeat2 web platform. In addition, we updated the web platform to improve descriptions, visual presentation and as mentioned a new functional module to explore the multicellular network.

Supplementary Note 8: User guide for ReHeat2 platform

ReHeat2 is an interactive web-based platform that enables the exploration of gene expression changes in HF through the integration of bulk and single-nucleus transcriptomics data. The platform, accessible at <https://www.saezlab.shinyapps.io/reheat2/>, allows users to access the built resource through two functional modules. The first module is a gene query that allows the examination of gene-specific expression patterns to investigate shifts in cell-type composition, and assess molecular reprogramming within individual cell types in response to HF. The second module is an interactive network that can be used to study a given cell-type pair of interest for their estimated communication events.

Users initiate their query in the first functional module by selecting genes of interest (Supplementary Fig. 14A). The platform then provides a comprehensive overview of their expression patterns in HF, integrating evidence from bulk and single-nucleus transcriptomic datasets (Supplementary Fig. 14B). Bulk transcriptomic data are first displayed as study-specific log fold changes (HF vs. NF), illustrating the variability in the direction of regulation across independent studies. To assess the statistical

significance of gene deregulation, the platform incorporates the consensus bulk HF signature, ranking genes based on their combined p-value across datasets. This allows users to evaluate the strength of association between their queried genes and HF across the large bulk cohort of patients. Beyond assessing differential expression in bulk tissue, ReHeat2 then provides insights into how transcriptional changes are coordinated at the cellular level. The next section examines whether the upregulation of a gene is primarily driven by shifts in cell-type composition or by intrinsic molecular deregulation. Cell-type-specific expression patterns are visualized to determine whether a gene is a marker of a particular cell type, which, in conjunction with compositional changes, can indicate regulation at the compositional level. To assess molecular regulation, the platform queries MCP1 to identify gene loadings per cell type, providing insight into the contribution of intrinsic transcriptional changes to overall gene expression shifts. Then, results are summarized in an automated text to aid with the interpretation of the plots. For example, a query of MXRA5, POSTN, and NPPA reveals that all three genes are significantly upregulated in bulk heart failure datasets. Single-nucleus data further resolve these changes at the cellular level, showing that NPPA is primarily upregulated within cardiomyocytes (molecular shift), while MXRA5 and POSTN are upregulated in fibroblasts, reflecting a combination of compositional and molecular shifts. By integrating these complementary analyses, ReHeat2 enables a nuanced interpretation of transcriptional regulation in HF, facilitating hypothesis generation.

The second functional module enables querying the multicellular network where users can explore specific cell type pairs and their estimated communication events (Supplementary Fig. 14C). The displayed network highlights the most significant edges, providing an overview of key interactions in failing or non-failing hearts. Selecting an edge of interest automatically generates a series of plots to estimate crucial communication events that may mediate the observed dependency in the network. The platform offers i) an overview of relevant ligand-receptor pairs, ii) ligand prioritization based on both expression scores and NicheNet regulatory potential, and iii) an analysis of the top downstream target genes potentially influenced by the ligand (Supplementary Fig. 14D). By studying these interactions, researchers can gain insights into the estimated molecular mechanisms driving cell type interactions in HF.

Thus, the ReHeat2 web platform enables a quick consultation of the combined evidence of the re-analyzed and combined 34 HF studies to facilitate hypothesis generation and translational insights.

Supplementary Figure 14. User guide for ReHeat2 web platform

- A. Cardiovascular researchers can query genes of interest to explore their expression patterns in heart failure (HF).
- B. ReHeat2 (Reference of the HEArt failure Transcriptome) is web platform that provides a user friendly query of single genes. Bulk transcriptomic analysis allows users to assess the statistical strength of a gene's association with HF. Study-specific log fold changes (HF vs. NF) illustrate variability across datasets, while a consensus HF signature ranks genes based on their combined p-value, providing an integrated measure of significance. Single-nucleus transcriptomics dissects whether a gene's deregulation is driven by changes in cell-type composition or by intrinsic molecular reprogramming, or both. Cell-type-specific expression patterns indicate whether a gene is a marker of a particular cell type, which, in combination with compositional shifts, suggests population-level regulation. Molecular regulation is further examined using MCP1 gene loadings, identifying cell-intrinsic transcriptional changes.
- C. Users can analyze cell-cell interactions in HF by querying cell type pairs to explore their dependencies and potential mediating ligands.
- D. The "Multicellular Network" tab highlights key edges in HF and non-failing heart networks. By selecting an edge of interest, users can generate ligand-receptor pair plots, ligand prioritization plots, and ligand-target gene visualizations.

Reviewer #2 (Remarks to the Author):

The manuscript presents a meta-analysis on the transcriptomic patterns of Heart failure (HF) and non-failing (NF) patients. By integrating multiple studies from both sn- and bulk- RNA sequencing profiles, this study depicts a HF patient map by analyzing the cell-type composition, cell-cell coordination & communication, multicellular programming, and labor division within fibroblasts.

In all, I find the topic of this study novel and meaningful. The study employs various computational techniques; however, the authors do not provide sufficient methodological details or adequately interpret the output results, making the manuscript difficult to follow. The manuscript would benefit from additional clarification and greater rigor. Specific comments are as follows:

Major comments:

1. The multicellular program (MCP) is one of the core concepts throughout this study. Although it has been briefly introduced with proper citation, I find it necessary to clearly describe this concept using both intuitive illustration and rigorous definitions. Otherwise, readers may not understand what this means and how it is obtained without prior knowledge on the author's previous work MOFA.

We thank the reviewer for this observation and we agree that more details across the introduction and results section will help the readers to better follow our multicellular integration work and the added value of our analyses.

We have modified the Introduction:

Understanding multicellular cooperation is critical because it ensures proper functioning of the heart and to what extent distinct pathological processes trigger similar multicellular responses in patients is not known. Emerging computational methods that leverage group latent variable models enable the inference of multicellular programs—structured patterns of coordinated gene expression changes across multiple cell types. Unlike traditional single-cell analyses that focus on individual cell types, multicellular programs capture both cell-type-specific and multicellular processes simultaneously, offering a broader systems-level perspective. Given the availability of single-cell data from cardiomyopathies, applying multicellular integration methods presents an opportunity to move beyond cell-type classification and incorporate a tissue-centric perspective. Moreover, this approach allows for the reinterpretation of bulk transcriptomic data beyond cellular composition, complementing single-cell studies that often have limited patient cohorts. Thus, multicellular integration offers a powerful framework to study multicellular processes in heart disease using the available omics datasets.

And the Results section 2.2.2:

Next, we explored whether the observed differences in tissue composition among patient groups influenced the comparability of the gene expression profiles of the cell-types in tissue samples from failing and non-failing hearts across single-nucleus studies. We reasoned that, since cells in tissues function as collectives, the molecular state of a tissue sample could be represented by coordinated transcriptional events across multiple cell types where gene expression changes of one cell-type relates to the changes of other cell-types, referred to here as multicellular programs (MCPs). To infer MCPs that describe patient variability within each study, we applied multicellular factor analysis²⁷. Briefly, this method takes pseudobulk gene expression matrices from each cell-type and infers a latent space that decomposes tissue sample variability. Each latent component defines an MCP, capturing coordinated gene expression changes across cell types. For each MCP, we obtained patient-specific activation scores and gene weights across cell types, enabling the interpretation of multicellular

transcriptional states (Supplementary Figure 5B). The multicellular programs of individual studies in the SN-core collection captured variance related to HF and other clinical covariates, clearly separating failing from non-failing hearts. These results suggested the presence of multicellular programs that reflect HF-associated tissue remodeling within each study (Supplementary Note, Figure 2G, Supplementary Figure 5C-E).

In addition, we added a panel in Supplementary Figure 5, that clarifies our multicellular approach:

B. Multicellular factor analysis leverages single-cell data from multiple samples to infer multicellular programs (MCPs). First, pseudobulk expression matrices are generated for each cell type, creating a multi-view representation of the data across samples, where each view reflects the aggregated gene expression profile of a specific cell type. Next, group factor analysis, as implemented in MOFA, is applied to infer a latent space that captures sample variability while integrating information across all cell types simultaneously. Each latent component represents an MCP, describing coordinated gene expression changes across cell types, which can reflect shared or cell-type-specific processes.

2. In this study, cell annotation for single-cell datasets is performed individually for each dataset. However, this approach may make it more challenging to compare cell types across studies. It is not clear how the cell types are aligned across different studies, as they might use different standards of cell type annotation and different sets of cell type terminology.

We apologize for the lack of clarity of our approach. In this integration of datasets we relied on the annotations provided by individual studies and we mapped them to a unique ontology, which means that we manually aligned the labels using regular expressions (e.g. fibs and Fibroblasts were mapped to a unique Fibs label). As we mentioned in our methods section:

4.2 Processing of single-nucleus studies.

[...]To ensure the comparability of the analysis across atlases, we defined a heart cell ontology that included the following cell types: Cardiomyocytes (CMs), fibroblasts (Fibs), endothelial cells (Endos), pericytes (PCs), vascular smooth muscle cells (vSMCs), myeloid and lymphoid cells. Regular expressions over original cell annotations provided by the selected studies were performed to align each dataset to our proposed ontology. Unannotated cells were discarded. Single-nucleus studies were transformed into collections of pseudobulk expression profiles by summing up the counts of all cells belonging to each of the cell types defined in our ontology for each sample. Quality metrics including the number of cells used, the total number of reads, and number of genes with available counts were calculated for each pseudobulk sample.

To estimate the agreement between the new assigned cell annotations, we compared the overlap of marker genes of the seven cell-types estimated from each study independently using Jaccard Indexes. Marker genes of each cell type per study were calculated using differential expression analysis between the pseudobulk expression profiles across samples of one cell-type versus the rest using edgeR v4.0.2 with default parameters of gene filtering. Genes with a log-fold-change over 2 and a false discovery rate (FDR) lower than 0.01 were considered cell-type markers.

The results of the comparison of cell-type markers were presented in Supplementary Figure 3D and suggest that the mapping of original annotations to the harmonized ontology group cells into the same lineages.

Supplementary Figure 3D Distribution of Jaccard Indices representing the pairwise similarities between sets of gene expression markers of cell-types (Methods). Each dot represents a comparison between the markers of cell-type X with markers of cell-type Y for every combination of SN-core studies.

Another alternative approach is to integrate multiple datasets, identify unified clusters and annotate these clusters. The authors need to justify the choice of their annotation strategy.

Since our analysis relies on pseudobulk representations of major lineages for dataset comparison and MCP construction, we chose an ontology-based harmonization of cell labels to ensure consistent grouping of cells within their respective lineages. In our previous work (Ramirez Flores *et al*, 2023) we found that this approach is sufficient for dataset comparability and robust MCP inference, providing a biologically meaningful framework for studying multicellular processes. The cell-marker comparisons across independent studies (Supplementary Figure 3D) provide confidence that this strategy effectively grouped cells into their corresponding lineages. Single-cell integration—which aims to classify, name, and organize cells into taxonomies across studies—is a parallel effort to our approach, which focuses on patient-level multicellular processes rather than individual cell identities. At the same, for cell-centric analyses full integration, as suggested by the reviewer, is necessary, which is why we performed a dedicated integration for fibroblast-specific analyses, ensuring a unified annotation across studies.

We added our assumptions in the results section (2.1 Study curation for the creation of the reference of the heart failure transcriptome) to clarify our choice:

[...]To facilitate the comparison of all studies we aligned meta data and cell type annotations, by mapping the original labels provided by each study to a harmonized vocabulary covering seven major cell-types: Cardiomyocytes (CM), fibroblasts (Fib), pericytes (PC), and endothelial (Endo), vascular

*smooth muscle (vSMCs), myeloid, and lymphoid cells. This mapping was performed using regular expressions to standardize cell-type naming across datasets. Rather than performing full single-cell integration, we opted for ontology-based harmonization, which ensures biologically meaningful alignment of broad cell types that is sufficient for dataset comparability and MCP inference*²⁷.

3. The manuscript uses technical jargon that may limit accessibility to a broader audience. For example, “However, transcriptional directionality was conserved, as shown by cross-study enrichment analysis (upregulated mean enrichment score 0.58, downregulated mean enrichment score -0.54, Supplementary Figure 2B),” what does the upregulated/downregulated mean enrichment score mean? How strong are these scores? What is the statistical significance (e.g., p-value, FDR) of these enrichment scores?

We apologise for the misunderstanding and the technical jargon. We have edited the manuscript to make the analysis more accessible. This includes a scheme describing the inference of multicellular programs (Supplementary Figure 5B), as well as an extended section in the introduction, a scheme explaining the generation of the cell type network and communication analysis, and a more detailed methods section.

Since the methodology regarding the bulk analysis was in part already published in our previous work, we kept the descriptions and discussions short in the current manuscript with reference to our previous work. Briefly, the distribution we are referring to consists of all pairwise combinations of enrichment of up and downregulated genes of one study to the gene level statistics (i.e. t-value gene ranking) of another study. Thus, the enrichment score indicates whether differentially expressed genes (for which we found little overlap) are nonetheless regulated in the same direction and therefore tend to be enriched on the extremes of the t-value ranking of the other study, as represented by high enrichment scores. To better assess the significance of the enrichment scores, we now included the visualisation of an adjusted p-value in the heatmap (Supplementary Figure 2B).

To clarify this part of the analysis we added a more detailed explanation to the results and methods section.

From **Results 2.2.1**:

[...] We compared the bulk studies in a pairwise manner to assess their agreement on gene expression changes reported in HF (methods) and we found that the overlap of differentially expressed genes of the new studies was low (mean Jaccard index 0.055, Supplementary Figure 2A). Cross-study enrichment analysis showed that upregulated genes in one study tended to be enriched at the top of the gene-level ranking (mean enrichment score = 0.58), while downregulated genes were enriched at the bottom of another study (mean enrichment score = -0.54, Supplementary Figure 2B). This indicated that despite the low overlap in differentially expressed genes, their transcriptional directionality was conserved. Consistent with this, pairwise study classification yielded high accuracy (mean AUROC = 0.89, Figure 2B; Methods), confirming that studies captured the same transcriptional patterns in HF, in agreement with our earlier findings (Ramirez Flores et al, 2021). These results supported the inclusion of new studies to refine the consensus signature through a gene-level meta-analysis (Figure 2C).

From **Methods 4.4**:

[...] Differential gene expression analysis, study comparison, classification and gene expression meta analysis was performed as described in our previous work (Ramirez Flores et al, 2021). Briefly, we

assessed the ability of individual studies to classify samples from other studies by defining a disease score. The disease score linearly combines a sample's gene expression values with a disease pattern from an independent reference study, captured by *t*-values from differential expression analysis. This estimates how closely a sample's expression profile aligns with the reference phenotype, emphasizing coordinated gene regulation over mean expression changes. To assess whether classification performance reflected consistent transcriptional regulation, we split the top 500 differentially expressed genes (sorted by *p*-value) into up- and downregulated sets and tested their enrichment in other studies' gene-level statistics via GSEA, with genes ranked by *t*-value. The consensus signature was derived by combining the Benjamin–Hochberg corrected *p*-values of the differential expression analysis for all genes that were measured in at least 10 data sets using a Fisher's combined probability test. The updated consensus was compared to the previous ranking by Jaccard indices.

Updated Supplemental Figure 2B.

B Enrichment score (ES) of the top 500 differentially expressed genes between failing and non-failing hearts of each study in the sorted gene-level statistics list of each other study. Colored study names are new studies added. Stars indicate an adjusted *p*-value < .05.

Also, how is the co-expression network defined in Section 2.4? It is confusing that the authors claimed “a consistency between the number of ligand-receptor interaction and co-expression network” but “a weak correlation between ligand-receptor coexpression and gene expression coordination”.

We apologise for the misunderstanding, we agree that the presentation of the co-expression network and the ligand-receptor analyses requires an extension to clarify different questions, similarly as the ones posted by Reviewer 1's comment 1.2. The network is built as mentioned in the Methods section “4.12 Multicellular coordination networks”. Briefly, the objective is to reconstruct from the information of a multicellular program a cell to cell network that describes to what extent the gene expression of the

population of a given cell-type can explain the gene expression of the population of another cell-type. This task can be formulated as a predictive model, where the target variable to predict is the expression of a cell-type in a patient using the rest of the cell-types as predictors. To avoid the use of a larger set of predictive features, we reduce both the expression of the target and predictor cell-types using enrichment scores. The model fit and coefficients are then used to define the weights of the network. We noticed that the use of different terms complicates the understanding of our analyses, so we have defined this network as a “multicellular coordination network” in the text and used it consistently across sections.

When we wrote about the consistency between the multicellular coordination network and the number of ligand-receptor pairs between cell-types we referred to the top cellular interactions we observed: fibroblasts regulating cardiomyocytes, and myeloid regulating endothelial cells. For both of these pairs we quantified a large number of ligands of the sender cell with potential interactions with receiver’s receptors and a large contribution of the sender cell in the prediction of the receiver cell in the coordination network. However, overall this pattern was not observed in all pairs of cells and interactions.

We have modified the text in results section **2.4 Blueprint of multicellular processes and cell-cell communication in heart failure** to clarify this misunderstanding:

[...]To investigate potential mechanisms of cell-cell communication underlying the failing-heart co-expression network captured by MCP1, we inferred ligand-receptor interactions across cell types (Supplementary Figure 7C, Supplementary Figure 8C). We observed a modest correlation between the number of potential ligand-receptor pairs in a cell type pair and their predictive importance in the multicellular coordination network (Spearman correlation = 0.33, p-value = 0.034, Figure 3F, Supplementary Figure 8D), suggesting that a minimal set of communication interactions could drive multicellular coordination (Figure 3F). The highest number of ligand-receptor interactions in HF was observed between fibroblasts and cardiomyocytes, as well as between myeloid and endothelial cells, which had, in addition, high predictive weights in the multicellular coordination network.

In addition, to facilitate the understanding of these analyses we have made schematics representing the framework to reconstruct communication mechanisms from multicellular programs that complement the descriptions of the models in the results section.

Supplementary Figure 7. Framework to infer the blueprint of coordination from multicellular programs and potential cell-cell communication.

- Given a multicellular program, the activation of cell-type specific components can be inferred using enrichment methods.
- The co-expression of the cell-type specific components of a multicellular program can then be represented as a network, here referred to as multicellular coordination network. In this network, each edge represents to what extent the expression profile of a given cell-type can predict the gene expression of the other. To build the network we fitted linear models where the target variable was the enrichment score of one cell-type and the predictors were the enrichment scores of all other cell-types. We used the model fit (R^2) and the coefficient estimated for each predictor to estimate the weight or importance of each edge. We kept all edges with weights larger or equal to 0.2.
- For selected important edges (i.e. with high weights), we used LIANA+ to select from the multicellular program represented in the network, all potential interacting pairs of ligand and receptors, taking into account the edge directionality and feature weight (i.e. ligands and receptors from the predictor and predicted cell-type, respectively). Ligand-receptor pairs with coherent signs of feature weights (i.e. the feature weights of both the ligand and receptor were all positive or negative) then are selected as relevant interactions. Ligands from these interactions and the expression profile of the predicted cell-type are then used as input for NicheNet.

4. The authors need to provide a rigorous description of how the projection of new dataset into the patient map. For example, what's the formula to calculate the MCP coordinates for a new dataset, given the current patient map?

We apologize for not properly describing the projection task, we have adapted the methods section to address this:

4.15 Study projection of supporting studies

Supporting single-cell studies were projected into the MCP latent space, estimated from the core studies using the gene weights of the MCPs. To project a new study we computed from the reference dataset the Moore-Penrose generalized inverse (Penrose and Todd, 1955) of the concatenated feature weights across cell-type views and multiplied it by the multi-view data of a test patient cohort. The inverse matrix of the feature weights was calculated using the `ginv()` function from MASS v7.3-57. The multi-view data of the test patient cohort was centered, where each feature had a mean equal to 0.

5. How to explain the low overlap of the differentially expressed genes among the bulk studies?

As discussed in our previous publication describing the integration of bulk data (Ramirez Flores *et al*, 2021), we have hypothesized that the low overlap between the top differentially expressed genes across studies may arise from differences in the statistical sampling of each patient cohort. Both the sample size and the non-probabilistic sampling affect the statistical power of the differential expression test that leads to distinct distribution of p-values and rankings (Meng, 2018). For the same reason, we focus on the consistency of gene regulation (Supplementary Figure 2).

We have updated the results section to mention this:

2.2.1 Comparison and meta-analysis of the core bulk transcriptomics studies

We compared the bulk studies in a pairwise manner to assess their agreement on gene expression changes reported in HF (methods) and we found that the overlap of differentially expressed genes of the new studies was low (mean Jaccard index 0.055, Supplementary Figure 2A). This low overlap aligned with our previous work¹⁶ showing that variation in cohort size and non-probabilistic sampling design affects the statistical power of differential expression testing and the rankings of p-values and t statistics. However, cross-study enrichment analysis showed that upregulated genes in one study tended to be enriched at the top of the gene-level ranking (mean enrichment score = 0.58), while downregulated genes were enriched at the bottom of another study (mean enrichment score = -0.54, Supplementary Figure 2B). This indicated that despite the low overlap in differentially expressed genes, their transcriptional directionality was conserved.

6. In detail, how was NicheNet performed when the input contained multiple snRNA-seq datasets from multiple samples? How does the cell-cell communication pattern vary across different studies within the HF group?

We employed NicheNet to interpret the derived HF-associated multicellular program (MCP1). Rather than running NicheNet separately on each dataset, we leveraged the meta-analyzed MCP1, which captures a conserved and generalizable HF signal across studies. Comparisons of the HF signal across datasets (Figure 2H), where we tested whether the transcriptional response to HF across cell-types of one study could classify HF patients in the other datasets, indicated a high degree of conservation and support our joint-modeling approach to extract conserved multicellular programs. Similarly, these results support the assumption that communication events, which are inferred from these expression changes, should exhibit a similar pattern. Thus, we performed communication analysis only after identifying the common HF signature in the form of MCP1. MCP1 provided gene loadings per cell type, representing linear feature weights that served as input for NicheNet. In more detail, we used NicheNet to identify ligands from a given cell type A that could explain the target genes (i.e., genes with extreme gene weights) of a cell type B. We have clarified this in Method section 4.13.

We performed ligand-receptor analysis to aid the interpretation of the MCP1. Thus, as input for this analysis we used the gene loadings of the MCP1 to extract possible active ligand-receptor pairs, as well as ligand to target gene associations. First, we extracted curated ligand-receptor pairs with liana v0.1.13 R-package⁵⁰. To calculate ligand-receptor interactions we extracted the expressed ligand and receptor genes with an absolute gene loading >0.1 in MCP1 for each cell type pair, and summed both loadings to a ligand-receptor interaction score. To connect these ligands with possible downstream targets, we then used the updated Nichenet resource. In detail, for a given pair of cell type A and cell type B, we selected all ligands expressed in cell type A and predicted target genes as defined by the top 10% most extreme gene loadings of the MCP1 from cell type B using the Nichenet algorithm. Background genes were defined as the 30% genes with MCP1 loadings closest to 0. Then, corrected AUPR was calculated as implemented in nichenetr v.2.0.4. R-package and together with L-R interaction score used for ligand prioritization.

7. What are the mostly altered ligand-receptor pairs and ligand-target pairs between HF and NF samples? The visualization of the cell-cell communications is not intuitive enough (especially the ligand-receptor pairs). I suggest using chord or circled diagrams connecting ligands and receptors to further visualize the dominant L-R pairs given the sender and receiver cell clusters.

We now added this insight, displaying the top 5% of ligand-receptor (LR) pairs across cell-types using their mean value of co-expression (MCP1 loadings) that we denoted as interaction score. We added this panel as Supplemental figure 8C to the manuscript. In addition, the full interaction table can also be accessed in Supplementary table 5.

Updated Supplemental Figure 8C

C. Heatmap showing top 5% of multicellular ligand-receptor interactions in MCP1. Numbers indicate the number of cell type pairs. Left and right heatmaps display cell type pair counts of ligands and receptors, respectively. Bar graph displays number cell type pairs colored by disease group.

8. The multicellular information network of NF hearts (like Figure 3E) is needed for a clear comparison.

We added the requested NF network as Supplementary Figure 8A. The comparison of edge strengths is highlighted in 8B where we find a trend of correlation between both networks. However, we tested and highlighted the edges with most divergent importances, yielding the emergence of the Fib to CM dependence.

Updated Supplemental Figure 8A

A. Multicellular information network of non-failing (NF) processes captured by MCP1, where each arrow describes how important the expression of a given cell-type is to predict the expression profile of another one (Methods). Predictive importances come from linear mixed models of cell-type signatures of MCP1.

Minor comments:

1. Figure 2D is not correctly cited in the main text.

We thank the reviewer for the attentive reading and cited Figure 2D in Results section 2.2.2:

To identify such a compositional disease signature of HF, we quantified and compared cell-type compositions across all tissue samples in the core SN studies (Figure 2D).

2. How is the term “comparable” defined in Section 2.2.2?

We assumed that if two populations of cells coming from different samples expressed similar marker genes, then both populations represented the same lineage. We believe that the changes done to the text as part of this reviewer’s second major comment should deal with this misunderstanding.

3. "Each data set was processed individually by applying common thresholds, filtering genes with at least 10 cells and cells with at least 200 genes and counts were log-normalized." It’s not clear what “common thresholds” mean here.

We apologize for the misunderstanding. By "common threshold," we intended to indicate that the same thresholds were applied to all data sets. We clarified this aspect now in the respective methods section

4.19 Single cell integration of fibroblasts :

[...]We integrated fibroblast single-cell data from the four core HF studies. To ensure that the single-nucleus data closely resembled the pseudobulk data and the derived MCP, we applied only minimal additional filtering. We retained cells with 200–2,500 detected genes and <1% mitochondrial gene content. Genes expressed in at least 10 cells were retained, and counts were log-normalized. Before integration, we removed the top 200 lineage markers from non-fibroblast cell types to reduce background contamination, after which highly variable genes were identified.

4. “We hypothesized that if two independent studies using the same technology captured similar molecular processes in HF, a disease classifier trained on one study’s data could predict the disease status of heart tissue samples in the other (Figure 2A). This approach allowed us to assess how well independent studies with different technical and clinical characteristics aligned at various scales.” It’s not clear what the different characteristics are for independent studies using the same technology.

Our analyses provided in Figure 1, Supplementary Figure 1, and Supplementary Figure 3 support the differences we highlight in that phrase of the manuscript. These figures show that despite the fact that these independent studies aimed at characterizing the molecular characteristics of HF at the single-cell level, they include diverse patient cohorts based on age, sex, and etiologies. In addition, they show expected technical differences including, different number of profiled cells, number of genes and sequencing depth. Given these differences, we performed our comparative analyses to evaluate their impact in identifying transcriptional signatures.

We have added a reference to the figures previously mentioned in this response to be more explicit about the technical differences we are referring to.

[...] This approach allowed us to assess how well independent studies with different technical (e.g. number of profiled cells, number of genes and sequencing depth) and clinical characteristics (Figure 1, Supplementary Figure 1) aligned at various scales.

References

Amrute JM, Lai L, Ma P, Koenig AL, Kamimoto K, Bredemeyer A, Shankar TS, Kuppe C, Kadyrov

- FF, Schulte LJ, *et al* (2023) Defining cardiac functional recovery in end-stage heart failure at single-cell resolution. *Nat Cardiovasc Res* 2: 399–416
- Amrute JM, Luo X, Penna V, Yang S, Yamawaki T, Hayat S, Bredemeyer A, Jung I-H, Kadyrov FF, Heo GS, *et al* (2024) Targeting immune-fibroblast cell communication in heart failure. *Nature* 635: 423–433
- Dimitrov D, Schäfer PSL, Farr E, Rodriguez-Mier P, Lobentanzer S, Badia-I-Mompel P, Dugourd A, Tanevski J, Ramirez Flores RO & Saez-Rodriguez J (2024) LIANA+ provides an all-in-one framework for cell-cell communication inference. *Nat Cell Biol* 26: 1613–1622
- Dimitrov D, Türei D, Garrido-Rodriguez M, Burmedi PL, Nagai JS, Boys C, Ramirez Flores RO, Kim H, Szalai B, Costa IG, *et al* (2022) Comparison of methods and resources for cell-cell communication inference from single-cell RNA-Seq data. *Nat Commun* 13: 3224
- Hoefl K, Schaefer GJL, Kim H, Schumacher D, Bleckwehl T, Long Q, Klinkhammer BM, Peisker F, Koch L, Nagai J, *et al* (2023) Platelet-instructed SPP1+ macrophages drive myofibroblast activation in fibrosis in a CXCL4-dependent manner. *Cell Rep* 42: 112131
- Meng X-L (2018) Statistical paradises and paradoxes in big data (I): Law of large populations, big data paradox, and the 2016 US presidential election. *Ann Appl Stat* 12: 685–726
- Ramirez Flores RO, Lanzer JD, Dimitrov D, Velten B & Saez-Rodriguez J (2023) Multicellular factor analysis of single-cell data for a tissue-centric understanding of disease. *eLife* 12
- Ramirez Flores RO, Lanzer JD, Holland CH, Leuschner F, Most P, Schultz J-H, Levinson RT & Saez-Rodriguez J (2021) Consensus Transcriptional Landscape of Human End-Stage Heart Failure. *J Am Heart Assoc* 10: e019667